# Fitness advantage of *Bacteroides thetaiotaomicron* capsular polysaccharide in the mouse gut depends on the resident microbiota

Daniel Hoces[1], Giorgia Greter[1], Markus Arnoldini[1], Melanie L Stäubli[2], Claudia Moresi[1], Anna Sintsova[2], Sara Berent[1], Isabel Kolinko[1], Florence Bansept[3†], Aurore Woller[3‡], Janine Häfliger[1§], Eric Martens[4], Wolf-Dietrich Hardt[2], Shinichi Sunagawa[2], Claude Loverdo[3], Emma Slack[1]*

[1]Institute of Food, Nutrition and Health, Department of Health Sciences and Technology, ETH Zurich, Zürich, Switzerland; [2]Institute of Microbiology, Department of Biology, ETH Zurich, Zurich, Switzerland; [3]Sorbonne Université, CNRS, Institut de Biologie Paris-Seine (IBPS), Laboratoire Jean Perrin (LJP), Paris, France; [4]Department of Microbiology and Immunology, University of Michigan Medical School, Ann Arbor, United States

*For correspondence:
emma.slack@hest.ethz.ch

Present address: †Department for Evolutionary Theory, Max Planck Institute for Evolutionary Biology, Plön, Germany; ‡Department of Molecular Cell Biology, Weizmann Institute of Science, Rehovot, Israel; §Klinik für Gastroenterologie und Hepatologie, University Hospital Zurich, Zurich, Switzerland

Competing interest: The authors declare that no competing interests exist.

**Abstract** Many microbiota-based therapeutics rely on our ability to introduce a microbe of choice into an already-colonized intestine. In this study, we used genetically barcoded *Bacteroides thetaiotaomicron* (*B. theta*) strains to quantify population bottlenecks experienced by a *B. theta* population during colonization of the mouse gut. As expected, this reveals an inverse relationship between microbiota complexity and the probability that an individual wildtype *B. theta* clone will colonize the gut. The polysaccharide capsule of *B. theta* is important for resistance against attacks from other bacteria, phage, and the host immune system, and correspondingly acapsular *B. theta* loses in competitive colonization against the wildtype strain. Surprisingly, the acapsular strain did not show a colonization defect in mice with a low-complexity microbiota, as we found that acapsular strains have an indistinguishable colonization probability to the wildtype strain on single-strain colonization. This discrepancy could be resolved by tracking in vivo growth dynamics of both strains: acapsular *B.theta* shows a longer lag phase in the gut lumen as well as a slightly slower net growth rate. Therefore, as long as there is no niche competitor for the acapsular strain, this has only a small influence on colonization probability. However, the presence of a strong niche competitor (i.e., wildtype *B. theta*, SPF microbiota) rapidly excludes the acapsular strain during competitive colonization. Correspondingly, the acapsular strain shows a similarly low colonization probability in the context of a co-colonization with the wildtype strain or a complete microbiota. In summary, neutral tagging and detailed analysis of bacterial growth kinetics can therefore quantify the mechanisms of colonization resistance in differently-colonized animals.

## Editor's evaluation

The authors have developed an innovative approach to analyze microbial population dynamics within a host and used this technique to address an important question – how the composition of the microbiota influences the intestinal colonization of an encapsulated vs unencapsulated version of an important commensal organism. The revisions add evidence to support their claims and make the approach more accessible.

## Introduction

From the moment that we first contact microbes at birth, we continuously encounter environmental and food-borne microbes. Whether such encounters are transient or will lead to long-term colonization is influenced by complex ecological interactions between the invading species and the existing consortium, as well as the host's dietary habits and the physiology of the intestine (*David et al., 2014*; *Wotzka et al., 2019*). A better understanding of the factors determining colonization efficiency is crucial in the development of microbiota engineering strategies (*Donia, 2015*; *Pham et al., 2017*; *Sheth et al., 2016*) and in the use of bacterial species as biosensors to probe microbiota function and stability (*Goodman et al., 2009*).

One established way of studying ecological processes within hosts is genetic barcode tagging of otherwise isogenic microbes. This has previously been used to study population dynamics of pathogens such as *Vibrio cholerae* or *Salmonella* Typhimurium within the infected host (*Abel et al., 2015*; *Vlazaki et al., 2019*). Based on barcode recovery and mathematical modeling, it has been possible to infer parameters such as growth, clearance, and migration rates (*Dybowski et al., 2015*; *Grant et al., 2008*; *Kaiser et al., 2014*; *Kaiser et al., 2013*), as well as the size of population bottlenecks imposed during colonization (*Abel et al., 2015*; *Li et al., 2013*), antibiotic treatment (*Vlazaki et al., 2020*), or immunity (*Coward et al., 2014*; *Hausmann et al., 2020*; *Lim et al., 2014*; *Maier et al., 2014*; *Moor et al., 2017*). Furthermore, combining neutral genetic barcodes with targeted mutant strains, this experimental tool can be used to mechanistically analyze the fitness effect of individual genes that regulate successful gut colonization or tissue invasion (*Di Martino et al., 2019*; *Nguyen et al., 2020*).

To study the dynamics of invasion of a novel microbiota member, we chose to use *Bacteroides thetaiotaomicron* (*B. theta*) as a model microbe. *B. theta* is a common commensal member of the human intestinal microbiota, and the availability of precise tools for genetically engineering it (*Lim et al., 2017*; *Mimee et al., 2015*; *Whitaker et al., 2017*) makes it a strong candidate for introducing novel functions into microbiomes. A common feature of *Bacteroides* species is the ability to use phase variation to modulate the expression of 3–10 capsular polysaccharide (CPS) operons, leading to the production of distinct capsule structures (*Porter and Martens, 2017*). *B. theta* strains lacking a capsule have been shown to engraft poorly in an existing microbiota when competing with CPS-expressing strains (*Martens et al., 2009*; *Porter et al., 2017*). However, the deletion of all capsule gene clusters is not expected to negatively influence the growth rate of *B. theta* per se (*Rogers et al., 2013*). Rather it can affect its survival on exposure to noxious stimuli, like bile acids, stomach acid, antimicrobial peptides, or phage (*Porter et al., 2017*; *Porter et al., 2020*). These characteristics of different *B. theta* strains make them a good model to test our ability to quantify population dynamics in vivo.

To quantify the processes determining success of *B. theta* colonization in the presence of different resident microbiota, we generated genetically barcoded *B. theta* strains able to produce capsular polysaccharides (wild type [WT]) or not (acapsular, with deletion of all eight capsular polysaccharide synthesis loci; *Porter et al., 2017*). The genetic barcodes were linked to an erythromycin resistance cassette to allow amplification of our barcodes by cultivation. Frequencies of barcoded strains in a sample can be easily determined by plating for CFU determination, recovering all colonies growing in the presence of the relevant antibiotic and quantifying relative barcode frequencies by quantitative PCR (qPCR). Combining these values gives CFU of each barcoded strain (*Grant et al., 2008*; *Lim et al., 2014*), which can be combined with simple mathematical models to estimate the probability of individual clones to colonize under different conditions. This revealed similar colonization success (i.e., encounter of similar population bottlenecks) between acapsular and WT *B. theta*, during colonization of mice carrying low-complexity microbiota (OligoMM12 and LCM microbiota). However, the probability of colonization dropped approximately 10-fold for WT strains and 100-fold for acapsular strains colonizing the gut of mice with a complex microbiota (specific pathogen free [SPF]). Despite similar fitness on single-colonization of low-complexity microbiota mice, the acapsular strain competed poorly against the WT strain in competitive co-colonization in the same setting. This apparent discrepancy between colonization probability and competitive fitness could be explained by a longer lag phase and very slightly reduced net growth rate of the acapsular *B. theta*. This gives WT *B. theta* a head-start to occupy most of the available niche, excluding acapsular strains. The same barcoding system can also be used to quantify the bottlenecks experienced by a steady-state *B. theta* population in the gut subjected to an acute inflammatory reaction. Therefore, neutral tagging and simple mathematical modeling can infer quantitative insights into the behavior of *B. theta* during gut colonization.

## Results

### Genetically barcoded *B. theta* strains to study within-host population dynamics

*B. theta* VPI-5482 can phase vary the expression of eight different capsular polysaccharides (WT). *B. theta Δcps* strain that cannot produce capsule was generated previously by sequentially deleting all CPS gene clusters (acapsular) (*Porter et al., 2017*). We first established neutrally tagged clones of these strains by inserting a genetic barcode linked to an antibiotic resistance cassette and a fluorescent protein gene into the genome using the previously described pNBU2 integration plasmid. Six barcode sequences, previously developed and validated for *Salmonella* (wild-type isogenic tagged strains [WITS]; *Grant et al., 2008*; *Maier et al., 2014*) were inserted adjacent to an erythromycin-resistance cassette, *ermG* (*Cheng et al., 2000*), and a GFP or mCherry fluorescent protein gene under a strong constitutive promoter. Untagged strains were generated by inserting pNBU2 carrying the *tetQb* tetracycline resistance cassette (*Nikolich et al., 1992*) and a GFP gene under the control of a strong constitutive promoter into the same integration site (*Figure 1A*, Key Resources Table). Fluorescent proteins (GFP or mCherry), expressed from a phage promoter with an optimized ribosome binding site, were included to permit later visualization of clones (*Wang et al., 2000*; *Whitaker et al., 2017*).

We validated a system to enrich barcoded strains from overall very low frequencies (*Figure 1B*) Samples were plated on BHIS agar with gentamycin to determine the total *B. theta* CFU and on BHIS agar containing the appropriate antibiotic (erythromycin or tetracycline) to determine the total barcoded *B. theta* CFU. Subsequently, all *B. theta* was washed from the plates, and genomic DNA extracted. The relative frequency of each barcode among the recovered colonies was obtained by qPCR using primer sets specific for each barcode. CFU of each individual barcoded strain can then be determined by simple multiplication. Serial dilution and recovery of barcoded strains in in vitro systems shows excellent resolution over five orders of magnitude (*Figure 1—figure supplement 1*).

### Genetic barcode tags do not strongly affect fitness of *B. theta* strains

A critical assumption of any analysis using genetically barcoded strains is that the chromosomal insertions, as well as the construction process, have not altered the fitness of the strains compared to the WT strain. Anaerobic growth in BHIS media was near-identical in all the barcoded *B. theta* strains (median doubling time ranged from 78 to 90 min) (*Figure 1C*). Correspondingly, the barcoded strains maintained their relative abundances, as evaluated by qPCR, when all barcode strains were mixed and grown overnight (*Figure 1—figure supplement 2A and B*). Whole-genome sequencing of the barcoded strains revealed a few synonymous and non-synonymous mutations, as would be expected for the construction process of individual strains (*Supplementary file 1*). However, none of the identified mutations is expected to have a major fitness effect, consistent with our observed data.

To test whether the tags confer a fitness effect upon colonization of a host, we colonized germ-free (GF) mice with a uniform mixture of $10^6$ CFUs of the untagged strain and each barcoded strain (*Figure 1D*). At this barcode abundance, stochastic loss of tags is highly unlikely. We compared the relative abundance of each barcode in the inoculum to that in the cecal content and feces after 48 hr of colonization. These experiments revealed small, random deviations in the distribution, consistent with uniform fitness (*Figure 1E*). Finally, as erythromycin- and tetracycline resistance were used to distinguish the barcoded and untagged *B. theta* strains, we also tested whether the antibiotic resistant cassettes alter competitive fitness. In culture, we found a minor competitive advantage of the tetracycline expressing strains over erythromycin only in the acapsular strain (*Figure 1—figure supplement 2C*). However, there was no significant competitive advantage of any antibiotic cassette after 2 days of colonization in GF mice (*Figure 1F*). As all barcoded strains carried the erythromycin resistance cassette, small fitness effects associated with the cassette will not affect competition between the barcoded strains. In competitive colonization, tetracycline and erythromycin resistances were reversed in two sets of experiments with similar results, and a simple model based on this data suggests a maximum error due to the antibiotic fitness effect of twofold. This is small compared to the relative competitive fitness of the acapsular and wildtype *B. theta* strains. Therefore, while absolute equivalent fitness is near-impossible to achieve in these systems, the error due to unintended fitness effects is within an acceptable range.

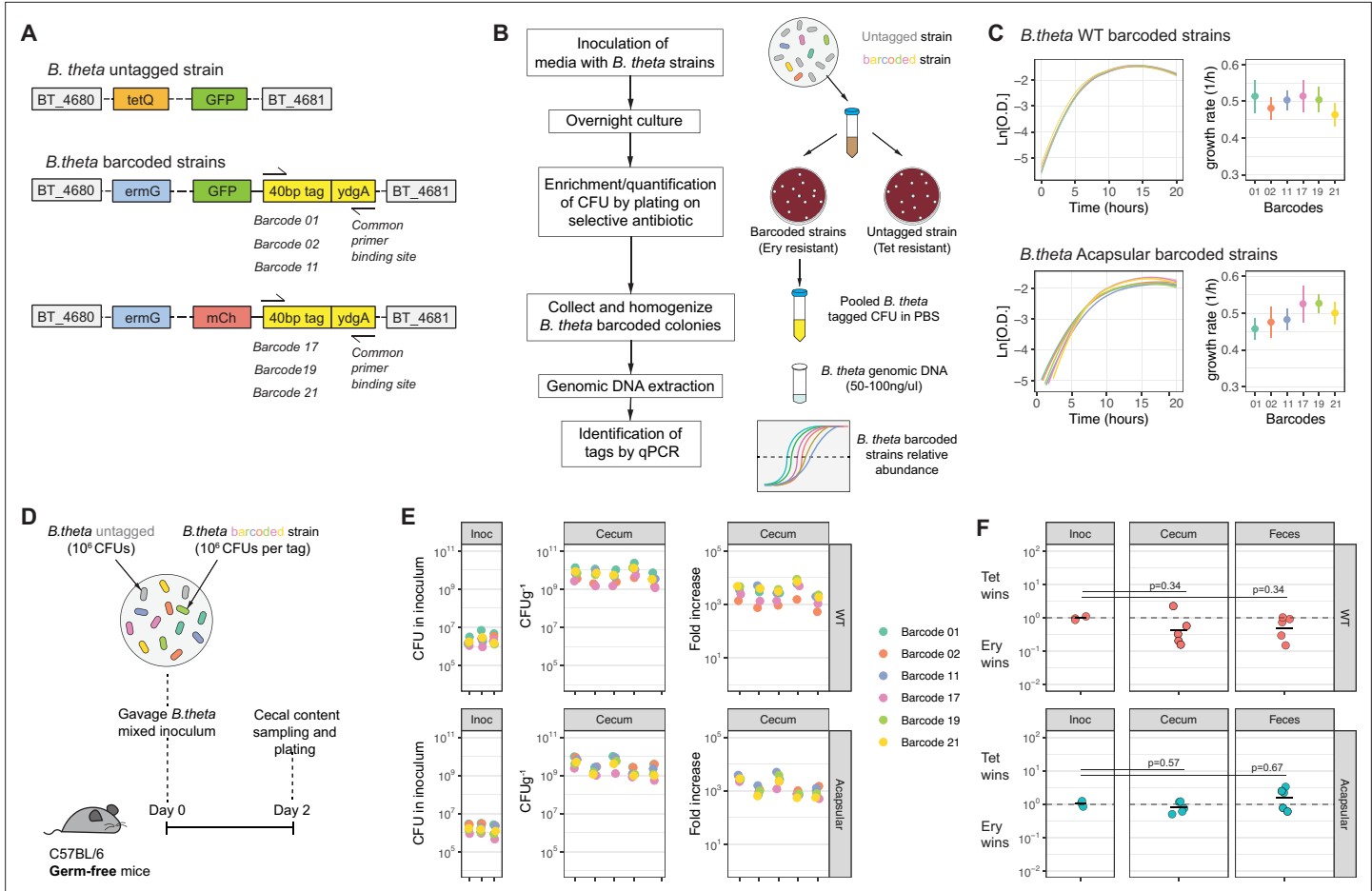

**Figure 1.** Barcoded *B. theta* strains have similar fitness for growing in vitro and in vivo. (**A**) Schematic representation of insertions in *B. theta* genome. The barcoded strains carried the barcode, erythromycin resistance cassette, and fluorescent protein in the genome. The untagged *B. theta* carried a tetracycline resistance cassette together with a fluorescent protein inserted at the same position in the genome. (**B**) Workflow for barcoded strain enrichment and quantification. (**C**) Growth curves of *B. theta* WT and acapsular barcoded strains in BHIS media (n = 3, replicative cultures per strain) and growth rates (1/hr, mean and 95% confidence interval) per barcode. (**D**) Experimental design of in vivo competitions to confirm equal fitness of the barcoded strains. Each strain (*B. theta* untagged and six *B. theta* barcoded) were mixed in an equal ratio (inoculum: 10⁶ CFU per strain). (**E**) Barcode distribution during colonization among six *B. theta* barcoded strains either WT or acapsular. Plots show distribution of barcodes in the inoculum, in cecal content of individual mice after 48 hr of colonization and fold increase of each barcode per mouse compared to the inoculum (n = 5 mice colonized). (**F**) Competitive index of tetracycline-resistant untagged strain (Tet) over the erythromycin-resistant barcoded strain (Ery) in vivo after 48 hr of colonization of *B. theta* WT or acapsular in cecal content and in feces (n = 2 replicative cultures in inoculum; n = 5 mice colonized). Points represent individual mice, and the horizontal line is the mean. p-Values were obtained by one-way ANOVA followed by Tukey's honest significance test. Data are included in *Figure 1—source data 1*.

The online version of this article includes the following source data and figure supplement(s) for figure 1:

**Source data 1.** Barcoded *B. theta* strains have similar fitness for growing in vitro and in vivo.

**Figure supplement 1.** Barcoded *B. theta* strains can be accurately detected in vitro.

**Figure supplement 2.** Barcoded *B. theta* strains have similar fitness for growing in vitro.

## Determining inoculum size of barcoded strains that yields maximal information upon *B. theta* colonization

We then applied the neutrally barcoded *B. theta* strains to estimate colonization probabilities. As invasion probabilities depend on the interaction with the resident microbiota (*Kurkjian et al., 2021*), we probed *B. theta* colonization in mice carrying three different communities: low-complexity microbiota (LCM) *Stecher et al., 2010*; Oligo Mouse Microbiota (OligoMM12) (*Brugiroux et al., 2016*), and SPF microbiota. These include two low-complexity microbiota models with a reduced set of strains (LCM: 8 strains and OligoMM12: 12 strains) and the closest model for complete microbiota in laboratory

mice (SPF: 12 families that include several species) (*Figure 2—figure supplement 1*). We evaluate the distribution of barcoded *B. theta* cells in cecum content at 48 hr after initial colonization, a time point shortly after the *B. theta* population in the cecum reaches carrying capacity (*Figure 2—figure supplement 2*).

Assuming that the change in relative abundance of tags before and after the colonization process is due to stochastic loss of *B. theta*, we formulated a simple 'initial' model that allows us to infer a per-cell colonization probability for *B. theta*. The model assumes that *B. theta* cells undergo random killing during their transition through the stomach and small intestine, that is, the population experiences an initial bottleneck event. Surviving cells arriving in the cecum start growing and the clonal progeny of these cells can be quantified at 48 hr post-colonization via a combination of plating and qPCR (*Figure 2A*). The number of cells of an individual barcoded clone in the inoculum, $n_0$, is low in our experiments. Correspondingly, the distribution of barcoded bacteria introduced into the stomach of each animal is better approximated by a Poisson distribution of mean $n_0$ than by a uniform distribution. The probability of losing a barcoded clone can be considered equivalent to the fraction of barcoded clones lost across all animals. Considering this early loss of clones as a binomial sampling process, we can express this probability as $e^{-\beta n_0}$, where $\beta$ is the colonization probability of an individual clone from the inoculum. $\beta$ can therefore be simply computed for animals all receiving an identical inoculum. To increase the power of our observations, we have also pooled data across multiple experiments with small deviations in $n_0$ by maximizing the likelihood of the experimental observations (*Figure 2—figure supplements 3 and 4*). A more complex calculation can be carried out using the variance of the barcoded population rather than defined loss/retention, a method that can take more information into account, although it is more complicated to execute (see 'Mathematical modeling').

It is important to note that if all tags are lost, or if all tags are recovered, only upper or lower bounds for $\beta$ can be estimated. Correspondingly, experiments where some, but not all, tags are lost from the final population yield maximum information. To find the optimal $n_0$ that leads to partial barcode loss in vivo, we titrated the barcoded strains into an untagged *B. theta* population to give $n_0$ values ranging from 18 to 26,666 CFU for each barcoded strain (*Figure 2—figure supplements 3 and 4*). This was carried out in the context of three different microbiota communities, and for both WT and acapsular *B. theta*. Recovery of total *B. theta* and CFUg⁻¹ of barcoded strains from the cecum was determined at 48 hr post-colonization. We used the Pielou evenness (*Pielou, 1967*) as a summary representation of the distributions of barcoded population (*Figure 2B*, *Figure 2—figure supplements 3 and 4*). The resulting $\beta$ estimates are most robust at an $n_0$ which results in approximately half of the tags being lost (see Appendix 1 'Supplementary methods'). In the case of LCM and OligoMM12 low-complexity microbiota mice, an $n_0$ of between 10 and 100 was optimal for both wild-type and acapsular *B. theta*, that is, 10–100 CFU of each barcoded strain was spiked into an inoculum of $10^7$ untagged *B. theta*. In the SPF mice, carrying a complex microbiota, an $n_0$ of around 500 was informative for WT, but 5000 CFU were needed of each acapsular barcoded strain.

Finally, we challenged the assumption that loss of barcoded clones was due to stochastic loss. Selective sweeps of a clone, or clones, that have acquired a beneficial mutation would also explain barcode loss. We therefore reisolated abundant barcoded *B. theta* strains from the 48 hr time point from previous experiments. These were used to assemble an inoculum in which some barcodes were represented by re-isolated strains and others by original ancestral strains. These were mixed at a high $n_0$ (approximately $2 \times 10^6$ CFU of each clone per inoculum) and used to colonize SPF mice. There was no consistent advantage of re-isolated strains over ancestral strains in colonizing the cecum at 48 hr post-colonization (*Figure 2—figure supplement 5A*), consistent with the absence of strongly beneficial mutations in surviving clones. To further confirm the assumption of equal stochastic loss of each barcode, we re-calculated $\beta$ for all experiments excluding each individual barcode in turn. Excluding data from any one barcode has no statistically significant effect on the estimates of $\beta$, as calculated using barcode loss (*Figure 2—figure supplement 5B*) or variance (*Figure 2—figure supplement 5C*).

## *B. theta* colonization probability in LCM, OligoMM12, and SPF mice

The resident microbiota composition in the mammalian gut is one of the main factors constraining the colonization of newly arriving species. This can happen through various mechanisms such as competition for nutrients (*Bäumler and Sperandio, 2016*; *Maier et al., 2013*), modification of the intestinal environment (*Cremer et al., 2017*), via direct suppression of the invaders by phages (*Almeida et al.,*

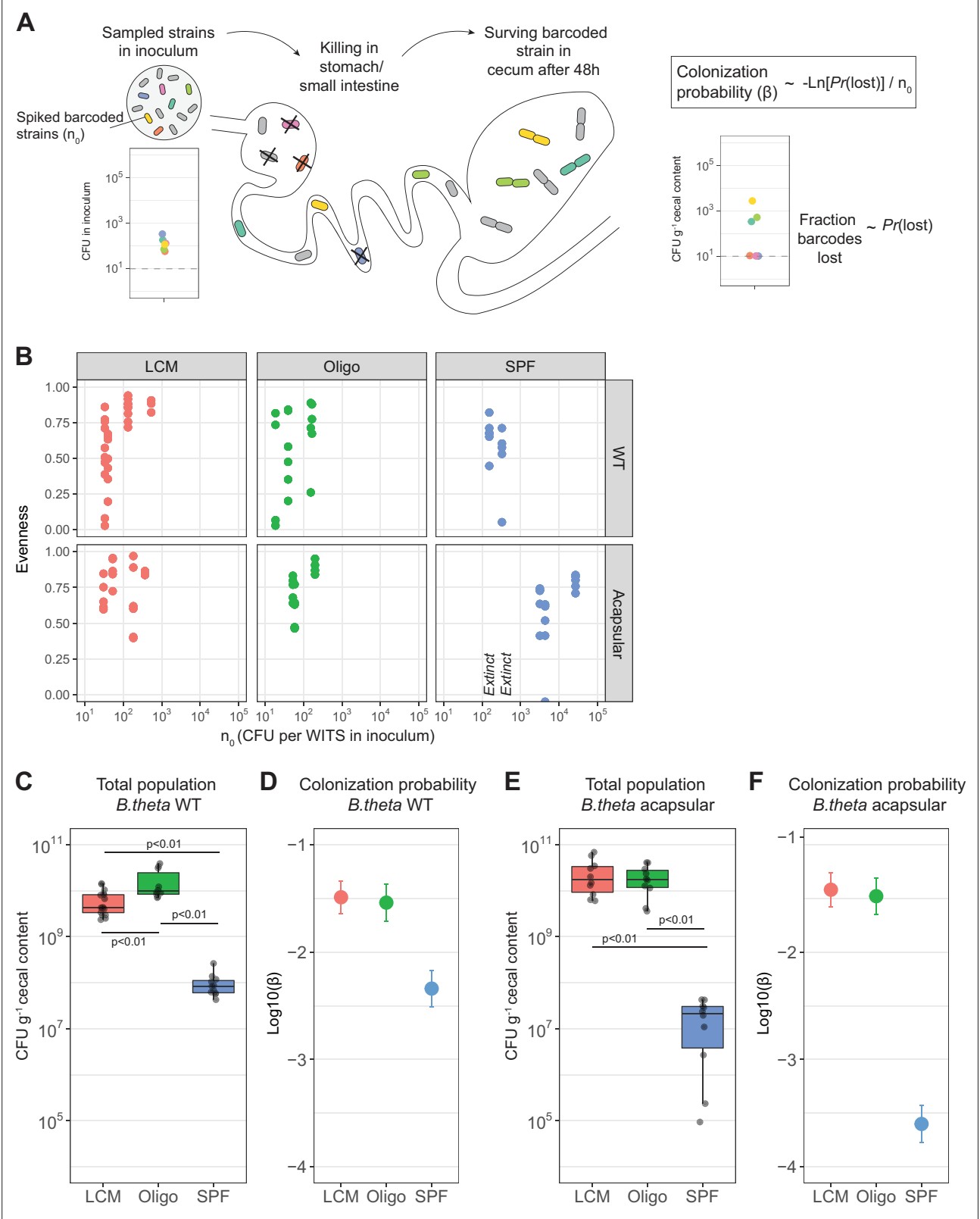

**Figure 2.** Colonization probability of *B. theta* strains in low-complexity microbiota (LCM), Oligo Mouse Microbiota (Oligo-MM12), and specific pathogen free (SPF) mice. (**A**) Schematic representation of experimental estimation of colonization probability. The untagged strain was tetracycline-resistant, and all barcoded strains were erythromycin-resistant. (**B**) Pielou's evenness vs. $n_0$. Pielou's evenness was estimated with a maximum possible value of $H_{max} =$ ln(6) for all data points (six barcoded strains). Each dot represents the evenness calculated per mouse. Values of $n_0$ for which all barcodes were lost, and

*Figure 2 continued on next page*

*Figure 2 continued*

therefore no evenness could be estimated, were marked with 'Extinct.' Total inoculum size was maintained at $10^7$ CFU. The exact inocula compositions are shown in *Figures 3 and 4*. (C, E) Total *B. theta* population in the cecum at 48 hr post-colonization for (C) WT and (E) acapsular strains. Points represent individual mice and boxplot quartiles provide summary statistics. p-Values were obtained by one-way ANOVA followed by Tukey's honest significance test. (D, F) Probability of colonization ($\beta$) in (D) WT and (F) acapsular in the cecum after 48 hr of colonization using the loss method. Circles represent the best estimate and vertical line the higher and lower bound of the 95% confidence interval. Estimation based on six barcodes times the number of mice (LCM = 17, OligoMM12 = 10, SPF = 11). See 'Methods' for parameter estimations. Data are included in *Figure 2—source data 1*.

The online version of this article includes the following source data and figure supplement(s) for figure 2:

**Source data 1.** Colonization probability of *B. theta* strains in low-complexity microbiota (LCM), Oligo Mouse Microbiota (Oligo-MM12), and specific pathogen free (SPF) mice.

**Figure supplement 1.** Fecal microbiota composition of different resident microbiotas.

**Figure supplement 2.** In vivo growth curve of *B. theta* WT.

**Figure supplement 3.** Distribution of *B. theta* WT barcoded strains in inoculum and after 48 hr of colonization.

**Figure supplement 4.** Distribution of *B. theta* acapsular barcoded strains in inoculum and after 48 hr of colonization.

**Figure supplement 5.** Barcoded strain persistence is not link to fitness advantage.

**Figure supplement 6.** Probability of colonization ($\beta$) using variance of barcoded strains before (inoculum) and after colonization (cecum).

**Figure supplement 7.** Acapsular *B. theta* strain is not targeted by IgA or phages in naïve specific pathogen free (SPF) mice.

**Figure supplement 8.** Time of colonization with specific pathogen free (SPF) microbiota in germ-free (GF) mice do not affect colonization probability of *B. theta* acapsular.

*2019*; *Barr et al., 2013*) or type VI secretion systems (*Chatzidaki-Livanis et al., 2016*; *Wexler et al., 2016*). Additionally, the microbiota stimulates host mucosal immunity and influences intestinal physiology (*Zheng et al., 2020*). *B. theta* loads in the cecal content at 48 hr post-inoculation were similar in LCM and OligoMM12 mice, but significantly lower in SPF mice (*Figure 2C*). The colonization probability, $\beta$, of barcoded *B. theta* WT strains, calculated using loss or variance methods, was also lower in SPF mice (Log10($\beta$, colonization probability) ± 2 standard deviations = –2.35 ± 0.14) compared to the two LCM (–1.50 ± 0.10; and Oligo, –1.54 ± 0.13) (*Figure 2D*, *Figure 2—figure supplement 6A*). Of note, while the relative size of the final population is 100-fold lower in SPF mice, and the relative colonization probability is only 10-fold lower than in animals with a low-complexity microbiota. This indicates that size of the open niche does not linearly translate into colonization probability, that is, the neutral tagging approach reveals information that cannot be simply gleaned from standard CFU determination.

As CPS are thought to play a crucial role in phage evasion/infection (*Porter et al., 2020*), immune evasion (*Fanning et al., 2012*; *Hsieh et al., 2020*; *Porter et al., 2017*), and protection from other environmental stressors, we expected to see a decreased colonization probability for acapsular *B. theta* strains in all microbiota backgrounds. Surprisingly, in LCM mice, the total population size of acapsular *B. theta* (*Figure 2E*) and the probability to colonize (Log10$\beta$: LCM, –1.43 ± 0.13; and OligoMM12, –1.49 ± 0.14) were not significantly different to the WT strain (*Figure 2F*, *Figure 2—figure supplement 6B*). There was therefore no measurable fitness benefit of CPS in gut colonization up to 48 hr post-inoculation in these settings. However, we observed a different scenario when we inoculated acapsular *B. theta* into mice carrying a SPF microbiota. Both the total population size of acapsular *B. theta* (*Figure 2E*) and the colonization probability (Log10$\beta$: SPF, –3.65 ± 0.13; *Figure 2F*) were tenfold lower compared to the WT strain, indicating a strong fitness benefit of CPS in the context of a more diverse microbiota. We could not definitively tie this increased clearance to any particular host or microbial mechanism: SPF mice do not have measurable IgA titers specific for acapsular *B. theta* (*Figure 2—figure supplement 7A*), nor could we identify lytic spots produced by phage specific for acapsular *B. theta* from the cecum content of SPF mice (*Figure 2—figure supplement 7B*). As microbiota-driven restriction of acapsular *B. theta* colonization is expected to establish very rapidly on recolonization of a GF mouse, but host-driven mechanisms such as upregulation of antibody responses may take several days to weeks, we compared the acapsular *B. theta* colonization probability in ex-GF mice that had been recolonized by rehousing with SPF mice for 2 days or for 2 weeks. Although short-term recolonization resulted in a larger *B. theta* population in the cecum than long-term re-colonization with an SPF microbiota (*Figure 2—figure supplement 8A*), the colonization probability was near-identical between the two groups (*Figure 2—figure supplement 8B*). Therefore, mechanisms

restricting colonization of acapsular *B. theta* in SPF mice are either direct microbial competition (e.g., via type VI secretion) and/or are very rapidly induced host mechanisms, or a combination of both.

## Competitive colonization with acapsular and WT *B. theta* reveals a role of CPS in OligoMM12-colonized mice

The absence of a decreased colonization probability for acapsular *B. theta* in LCM mice apparently conflicts with previous studies showing a competitive fitness defect of this strain (*Coyne et al., 2008*; *Porter et al., 2017*). We therefore carried out competitive colonizations with *B. theta* WT and acapsular strains in all microbiota backgrounds. Starting at a 1:1 ratio, we inoculated mice with erythromycin-resistant WT and tetracycline-resistant acapsular *B. theta* and quantified the cecal bacterial load 48 hr post-inoculation (*Figure 3—figure supplement 1A*). This reveals a gradient of competition with the WT winning over the acapsular strain, obtaining a competitive index (abundance of WT over acapsular *B. theta* at the end of the experiment) of approximately 20 in GF mice, 100 in LCM mice, and $10^4$ in SPF mice (*Figure 3A*). Therefore, the competitive fitness benefit of CPS increased in proportion to microbiota complexity, despite the fact that no difference on colonization probability of the WT and acapsular *B. theta* could be detected on single colonization.

To better understand the mechanisms generating a competitive disadvantage for the acapsular strains, we performed a competition experiment in the same microbiota backgrounds, but this time, using tetracycline-resistant *B. theta* WT and barcoded erythromycin-resistant acapsular *B. theta* strains. WT strain density (*Figure 3—figure supplement 1B*) and the average increase in the WT relative to the acapsular strains (after adjusting for the initial ratio in the inoculum, *Figure 3—figure supplement 1C*) were similar between the two experiments. Interestingly, the colonization probability $\beta$ was lower for the acapsular *B. theta* strain when co-colonizing with the WT strain than when colonizing alone (*Figure 3B*). This indicates that the competition with WT *B. theta* results in both a lower total population size and increased clonal extinction in the acapsular strains.

## Longer lag phase and higher clearance rate explains fitness defect of acapsular *B. theta* in competitive colonization

To understand how acapsular *B. theta* can have an indistinguishable colonization probability when inoculated alone, but a major fitness defect in competition with *B. theta* WT, we carried out time courses of feces collection to estimate the net growth rates (i.e., growth minus clearance) of both strains colonized individually in OligoMM12 mice (*Figure 3—figure supplement 2*). Using peak-to-trough analysis, we observed that growth rates of *B. theta* in feces and cecum is similar, and higher than all other OligoMM12 strains at 8 hr post-inoculation, as would be predicted for active growth (*Figure 3—figure supplement 3*). Longitudinal feces collection was therefore used as a proxy for large-intestinal colonization. Tracking *B. theta* CFUg$^{-1}$ in feces over 48 hr demonstrated that the net growth rate of acapsular *B. theta* is lower than that of WT (*Figure 3C*, WT: 0.50/hr and acapsular: 0.40/h, p<0.01). Interestingly, detectable exponential growth of acapsular strain starts around 4.5 hr later that for the WT (*Figure 3D*, WT: 3.2 hr and acapsular: 7.7 hr, p=0.02). As there is an inherent detection limit for CFU, as well as an intrinsic time delay due flow through the gastrointestinal tract this delay could be explained by (1) a classic lag phase (i.e., period of adaption before growth begins), (2) strongly increased killing of the acapsular *B. theta* during stomach and small intestinal transit, or (3) retention of acapsular *B. theta* in the non-growth-permissive small intestine. We could exclude differential retention of acapsular *B. theta* in the small intestine. Analysis of *B. theta* distribution in the small and large intestine at 8 hr post-colonization indicated that most *B. theta* had already arrived in the large intestine at this time point. There was no evidence of differential retention of acapsular and WT *B. theta* in the small intestine (*Figure 3—figure supplement 4*). We can also largely exclude killing prior to reaching the cecum as *B. theta* WT and acapsular clones have a very similar probability of colonization in single colonizations of OligoMM12 mice (*Figure 2* and figure supplements). For early killing to explain out-competition of the acapsular strain, it would be necessary that the presence of WT *B. theta* increases acapsular *B. theta* killing in the stomach or small intestine. As the bacteria are at low density during early infection, it is unlikely that they affect either each other or the host prior to arrival in the large intestine. In contrast, the delay in detectable growth of the acapsular strain is observed both competitive and single colonization of OligoMM12 mice, consistent with a classical lag phase. The magnitude of the delay to detection is similar on single and competitive colonization with

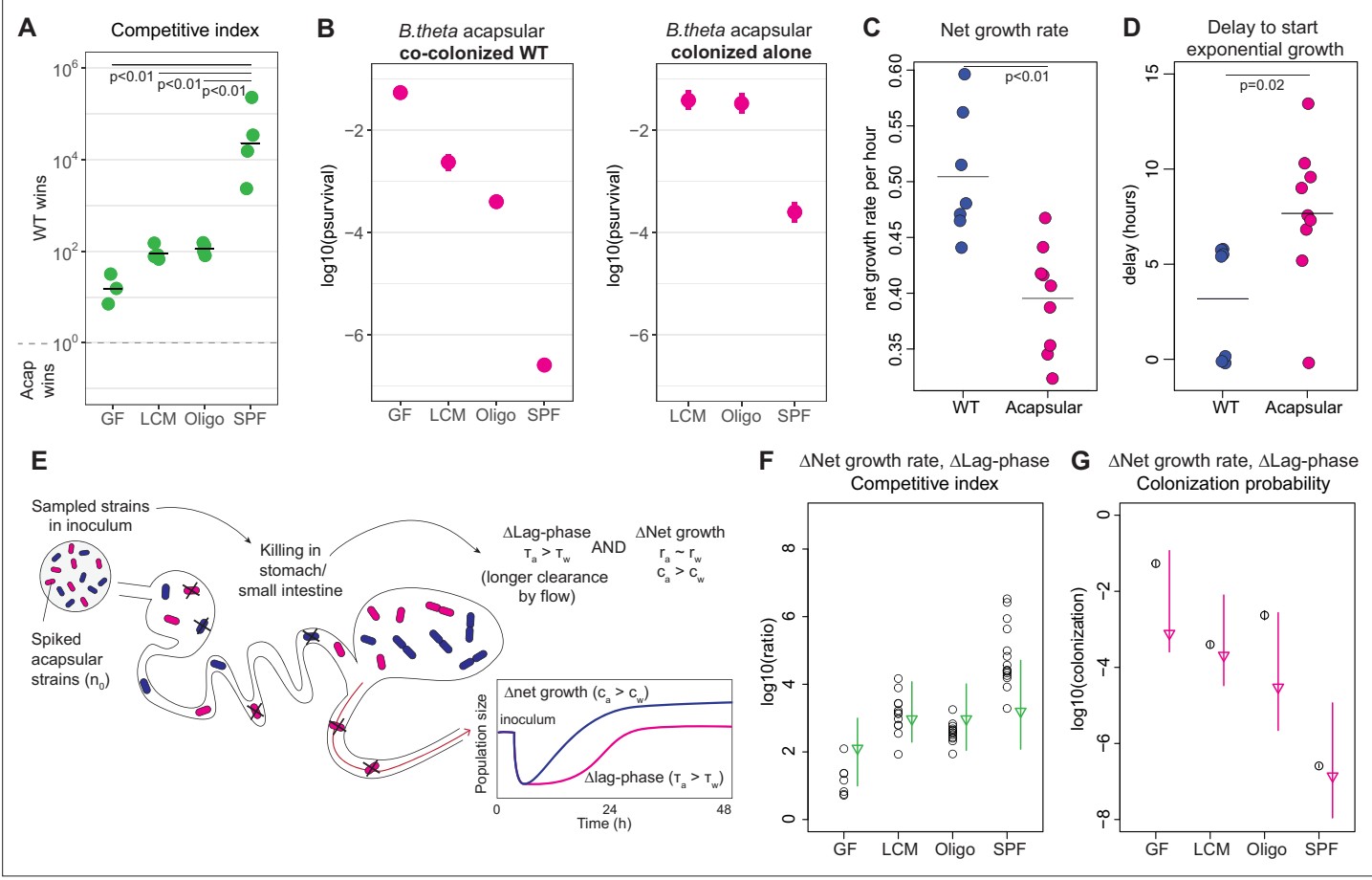

**Figure 3.** Competitive colonization with acapsular and WT strains. (**A**) Competitive index (ratio between WT over acapsular) in the cecum after 48 hr of colonization starting at a 1:1 ratio (inoculum: approximately $10^6$ CFU of each: erythromycin-resistant WT and tetracycline-resistant acapsular; germ-free [GF] = 3, low-complexity microbiota [LCM] = 4, Oligo Mouse Microbiota [OligoMM12] = 4, specific pathogen free [SPF] = 4). Points represent individual mice, and the horizontal line is the mean. p-Values were obtained by one-way ANOVA followed by Tukey's honest significance test. (**B**) (Left) Probability of colonization by the acapsular strain during co-colonization with the WT strain. Circles represent the best estimate and vertical line the higher and lower bound of the 95% confidence interval. Barcoded erythromycin-resistant acapsular strains were spiked into a WT untagged strain inoculum (inoculum: $10^7$ of untagged tetracycline-resistant *B. theta* WT and $n_0$ CFU of erythromycin-resistant barcoded *B. theta* acapsular adjusted to each microbiota background: $n_0^{GF} \sim 10$, $n_0^{LCM} \sim 10^3$, $n_0^{Oligo} \sim 10^3$, $n_0^{SPF} \sim 10^6$; number of mice per group: GF = 7, LCM = 13, OligoMM12=12, SPF = 16). (Right) Probability of colonization by the acapsular strain when colonizing alone. Graph generated using the same data as *Figure 2F*. (**C, D**) Estimation of (**C**) net growth rate and (**D**) and delay to start exponential growth (see *Figure 3—figure supplement 2* and Appendix 1 for fitting function; n = 7 for WT and n = 9 for acapsular). Points represent estimations of individual mice, and the horizontal line is the mean. p-Values were obtained by Welch *t*-test. (**E**) Schematic representation of competitive advantage of the WT over the acapsular *B. theta* having a similar initial probability of colonization of the cecum: difference in lag phase (mean time to growth commencement in acapsular ($\tau_a$) and WT ($\tau_w$)) and difference in net growth rate (growth rate in acapsular ($r_a$) and WT ($r_w$); clearance rate in acapsular ($c_a$) and WT ($c_w$)). Clearance can be due to both flow/loss in the fecal stream and death. (**F, G**) Estimation of the (**F**) competitive index and (**G**) colonization probability of the acapsular strain assuming a mean 4.5 hr difference in lag phase and the estimated difference in net growth rate between the WT and acapsular strains. Circles represent experimental data from (**B**) and *Figure 1C*. Triangles represent the best estimate and vertical line the higher and lower bound of the 95% confidence interval. See 'Methods' for parameter estimations. Data are included in *Figure 3—source data 1*.

The online version of this article includes the following source data and figure supplement(s) for figure 3:

**Source data 1.** Competitive colonization with acapsular and WT strains.

**Figure supplement 1.** *B. theta* acapsular is outcompeted by WT during co-colonization.

**Figure supplement 2.** Net growth of *B. theta* during colonization of Oligo Mouse Microbiota (OligoMM12) mice.

**Figure supplement 3.** Growth of *B. theta* and Oligo Mouse Microbiota (OligoMM12) in vivo.

**Figure supplement 4.** *B. theta* load across gut sections of Oligo Mouse Microbiota (OligoMM12) mice.

**Figure supplement 5.** Models for competitive index and colonization probability with fixed parameters.

WT *B. theta* (*Figure 3—figure supplement 2E and F*), consistent with this being an intrinsic feature of the acapsular *B. theta* strain. We therefore propose that acapsular *B. theta* exhibits a classical extended lag phase in vivo, likely due to a longer adaption period for the *acapsular B. theta* to growth in the gut environment.

To further explore this hypothesis, we extended our one-step colonization model to include both a difference in lag phase after arrival in the cecum and/or a difference in net growth rates (*Figure 3E*). Combining these additional variables generated a model that quite well recapitulates the expected competitive fitness (*Figure 3F*) and colonization probabilities (*Figure 3G*) (see 'Methods' for a brief description of the model and Appendix 1 'Supplementary methods' for the description of all parameters used). Running the same model based only on identical growth rates, but different lag phase produces a worse prediction of the competitive index, while omitting the lag-phase difference produces a similarly good fit (*Figure 3—figure supplement 5*). Therefore, the competitive fitness defect of the acapsular *B. theta* strains can be explained by a slightly slowed in vivo net growth rate, with a small contribution from an extended in vivo lag phase.

## Acute challenges modify *B. theta* population dynamics in vivo

Finally, as a proof of concept for neutral tagging in the study of established microbiota, we used our system to probe clonal extinction when an established *B. theta* population in the gut is challenged by two major environmental perturbations: (1) shifting from standard chow to a high-fat no-fiber diet (HFD) (*David et al., 2014*; *Wotzka et al., 2019*) and (2) infectious inflammation driven by *Salmonella* Typhimurium (Stm) (*Maier et al., 2014*). To exclude microbe–microbe interactions from the possible observed mechanisms, we monocolonized GF mice with a mixture of untagged and barcoded *B. theta* WT strains such that all tags were present with a roughly uniform distribution prior to intervention, that is, minimum loss, in the gut lumen prior to challenge (*Figure 4A*, *Figure 4—figure supplement 1*).

In the first set of challenges, after 4 days of colonization, we exposed mice to oral infection with $10^6$–$10^7$ CFUs of a Stm strain either attenuated (SL1344 *ΔssaV*, no SPI-2) or fully avirulent (SL1344 *ΔinvGΔsseD*, no SPI-1 or SPI-2). Despite similar Stm loads (*Figure 4—figure supplement 2A*), the attenuated strain induces moderate intestinal inflammation in GF mice while the fully avirulent strain does not induce visible gut inflammation (*Hapfelmeier et al., 2005*; *Stecher et al., 2005*; *Figure 4—figure supplement 2B*). *B. theta* populations were monitored in feces before the challenge was administered and 3 days after the challenge in feces and cecum. In line with the models presented above, cecum content values were used for inference of the bottleneck size.

After 3 days of infection with attenuated Stm, the total *B. theta* population was reduced approximately 100-fold in feces (*Figure 4B*). In addition, the probability of an established clone to survive this acute inflammation challenge was approximately 1 in 2000 in cecum (*Figure 4C*). This means that the initial estimated population of $10^{10}$ CFUg$^{-1}$*B. theta* in the cecum is reduced to an effective population of between $10^6$–$10^7$ clones, while maintaining a total population size of between $10^9$–$10^{10}$ CFUg$^{-1}$ (*Figure 4B*). Challenge with avirulent Stm has a limited effect on total population density of *B. theta* (*Figure 4B*); however, it still induces a bottleneck of approximately 1 in 250 in the cecum, potentially due to subclinical inflammation induced by colonization with a noninvasive *Enterobacteriaceae* (*Figure 4C*). Of note, we cannot formally exclude direct toxicity of *Salmonella* against *B. theta* based on these data. Rather around 90% of the *B. theta* clearance observed in virulent *Salmonella* infections can be attributed to the strong inflammatory response induced (*Figure 4—figure supplement 2B*).

In a second set of experiments, *B. theta* monocolonized mice were fed an HFD. Despite setting the initial barcoded population size in the cecum at between 100 and 1000 CFUg$^{-1}$ (*Figure 4—figure supplement 1*), HFD feeding did not significantly change the cecum population size (*Figure 4B*), nor did it increase the loss of barcoded *B. theta* (*Figure 4C*). To calculate our limit of sensitivity with this system, we estimated that detecting loss of one barcode in one of the tested mice would occur with a bottleneck population size of between $10^9$ and $10^{10}$ clones, that is., a population contraction of up to tenfold would be within the experimental noise of our measurements. Published reports indicate that *B. theta* is sensitive to bile acids (*Wotzka et al., 2019*), which should be abundantly induced by HFD feeding. The conclusion of the data is therefore rather than the current neutral tagging system is not sufficiently precise to detect the magnitude of population dynamics changes induced by HFD consumption in GF mice.

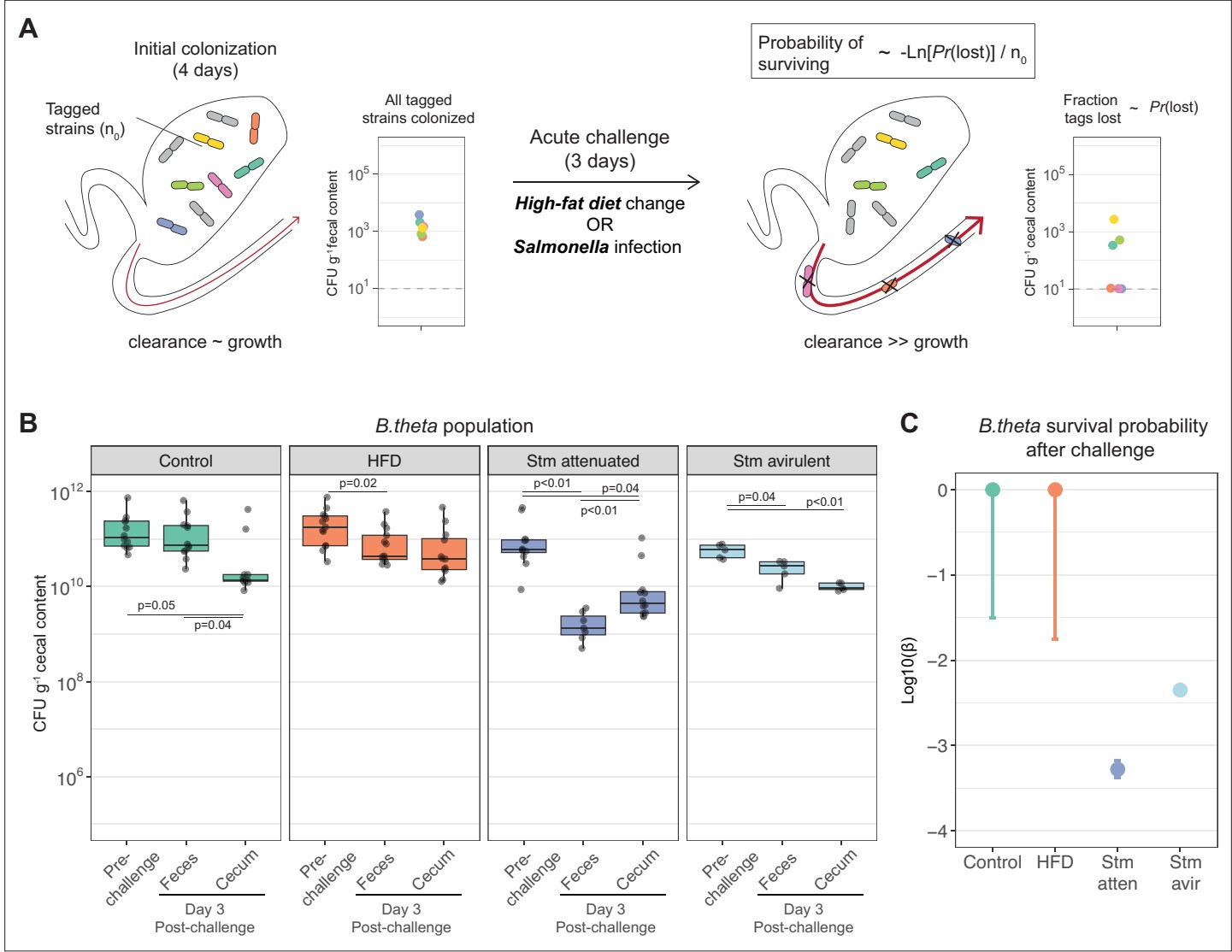

**Figure 4.** Acute challenges imposed a population bottleneck in the resident *B. theta* population. (**A**) Schematic representation of experimental estimation of colonization survival probability after acute challenges. (**B, C**) Germ-free (GF) mice were colonized with an inoculum of ~$10^9$ CFU untagged tetracycline-resistant *B. theta* WT spiked with barcoded erythromycin-resistant WT strains. The number of spiked CFU of each individual barcode per experiment ($n_0$) is described in *Figure 4—figure supplement 1*. (**B**) Population of *B. theta* in monocolonized ex-GF mice kept under standard chow (control) and during acute challenge with high-fat diet (HFD), infection with attenuated *Salmonella* (Stm) (Δ*ssaV*) or avirulent (Δ*ssaV*Δ*invG*) *Salmonella*. Points represent individual mice and boxplot quartiles provide summary statistics. p-Values were obtained by one-way ANOVA followed by Tukey's honest significance test. (**C**) Probability of surviving in the cecum after 3 days of the acute challenge. Estimation based on six barcodes timer the total number of mice (Control = 12, HFD = 13, Stm attenuated = 14, Stm avirulent = 5). Circle represent the best estimate and vertical line the higher and lower bound of the 95% confidence interval. Data are included in *Figure 3—source data 1*.

The online version of this article includes the following source data and figure supplement(s) for figure 4:

**Source data 1.** Acute challenges imposed a population bottleneck in the resident *B. theta* population.

**Figure supplement 1.** Distribution of *B. theta* WT barcoded strains in inoculum and after challenge.

**Figure supplement 2.** Inflammation induced by Stm attenuated challenge.

## Discussion

Understanding the different mechanistic factors determining a new species can invade into a resident gut microbiota is of considerable importance for combating mucosal infections (*Kreuzer and Hardt, 2020*; *Stecher, 2021*), but also for rationally introducing new functions into existing communities (*Cubillos-Ruiz et al., 2021*). These factors can include, among others, nutrient/energy availability,

environmental factors such as pH and flow/dilution rate (*Arnoldini et al., 2018*), and the presence of directly toxic or aggressive activities derived from the host (*Cullen et al., 2015*) or from other microbiota species (*García-Bayona and Comstock, 2018*), all of which can affect different microbes in different ways. Here, we condense all these mechanisms into three processes: factors affecting immigration rate (i.e., arrival into growth-permissive sites in the gut), factors affecting growth, and factors affecting clearance/death (*Hoces et al., 2020*). Combining in vivo experiments using genetically barcoded strains with mathematical modeling, we were able to empirically (e.g., net growth rates through plating) or deductively (e.g., probability of colonization) estimate the relative contribution of these three processes for colonization of *B. theta* under different conditions. In addition, we were able to quantify how fitness-relevant genetic changes in *B. theta* (production of capsular polysaccharide) affect colonization success.

When analyzing the effect of capsular polysaccharide expression on the process of colonizing mice with different resident microbiota, we found that the previously shown fitness disadvantage of acapsular strains (*Coyne et al., 2008*; *Porter et al., 2017*) depends on the microbiota context, rather than on host effects. The WT and the acapsular strains colonize mice with a relatively simple microbiota (LCM, Oligo) similarly well. However, in mice with more complex microbiota (SPF), *B. theta* can be outcompeted by the resident microbiota (*Lee et al., 2013*), and the acapsular strain engrafted significantly less well than the WT. While a possible explanation includes more robust intestinal immunity in fully colonized SPF mice (*Cullen et al., 2015*; *Hsieh et al., 2020*), it is also probable that expression of CPS is important for interaction with or protection against other microbes. Possible microbe-inflicted processes against which CPS can protect include microbe-on-microbe killing (*Chatzidaki-Livanis et al., 2016*; *Wexler et al., 2016*) and susceptibility to phages (*De Sordi et al., 2019*; *Porter et al., 2020*). Intriguingly, co-colonization experiments with mixed *B. theta* inoculums consisting of WT and acapsular strains recapitulate similar colonization probabilities for the acapsular strain as that observed in a complete SPF microbiota. As direct toxicity between the acapsular and WT strains can be largely excluded, this prompted us to examine growth kinetics of *B. theta* in the gut. Longitudinal analysis of fecal CFU densities demonstrated both a lower net growth rate in vivo (likely explained by a higher death rate rather than a lower replication rate) and a longer lag phase before commencing growing in the gut. Based on the data for competitive colonization in LCM mice, as well as the outcomes of our mathematical models, we can conclude that a small difference in net growth rate, combined with an extended lag phase, is sufficient to numerically explain the competitive fitness loss observed in numbers and colonization probability of acapsular *B. theta*.

The effect of two important challenges with known effects on resident commensals – high-fat feeding (*Wotzka et al., 2019*) and inflammation (*Maier et al., 2014*) – seems to depend on the ecological context in different ways. Feeding mice that are stably monocolonized with *B. theta* a fiberless HFD imposes a bottleneck that must represent less than a tenfold reduction in effective population size of the *B. theta* at 3 days post-treatment, even though this intervention has been shown to increase bile acid concentrations to levels that inhibit *B. theta* growth (*Wotzka et al., 2019*). It has been shown that *B. theta* rapidly evolves to adapt to dietary challenges in the context of a resident microbiota (*Dapa et al., 2022*). Therefore, the observed mild population bottleneck imposed by HFD feeding might only manifest if *B. theta* competes against other gut microbiota members, for example, that are more resistant to bile salts (e.g., *Escherichia coli*, see *Wotzka et al., 2019*). When infecting mice that are stably monocolonized with *B. theta* with *Salmonella*, we observed a larger decrease in *B. theta* clonal survival probability, which is consistent with the sensitivity of commensal species to gut inflammation (*Stecher et al., 2007*). Consistent with inflammation driving the main part of this phenomenon, we see less of a bottleneck when infected mice with a SPI1/2 double-mutant avirulent *Salmonella* that does not cause clinically overt inflammation in the gut. Interestingly, inflammation also causes a population bottleneck in the infecting *Salmonella* population (*Maier et al., 2014*), but this is less pronounced than the one we observe for *B. theta*, and the total population size of *Salmonella* rapidly recovers after this bottleneck. Therefore, the rapid killing/clearance of gut luminal *B. theta* seems to be representative of microbiota suppression that underlies the loss of colonization resistance in *Salmonella*-induced colitis.

Our mathematical models are based on certain assumptions that are useful to simplify calculations but always risk introducing bias. Most notably, we have made estimates for a single population of bacteria that has a constant growth and clearance rate. Necessarily the reality is more complex than

this – the nutrient profile and motility of the intestine will vary with circadian rhythm. Also, previous work has demonstrated that particular CPS-expressing clones may have an advantage in colonizing the dense mucus layers of the distal colon (*Donaldson et al., 2018*) although LCM studies suggest that at the population level *B. theta* grows at a similar rate in the mucus layer and lumen (*Li et al., 2015*). Nevertheless, recognizing these limitations, our estimates of colonization probability, growth, and clearance rates still give a good overview of the harsh processes with strong effects on the *total* intestinal *B. theta* population. It is also interesting to compare the neutral tagging approach to sequencing-based methods for growth rate estimation. It should be noted that these techniques give different information: peak-to-trough replication rate analysis reveals the growth rate at the time of measurement, whereas neutral tagging typically reveals population dynamics parameters averaged over much larger time spans. The approaches also have different limitations. While sequencing requires that the strain of interest is >0.1% of the total microbiota in order to generate sufficient confidence in the reads with reasonable sequencing runs, neutral tags can be used to examine very small populations. However, far more factors influence the interpretation of neutral tagging experiments (bottlenecks, clearance rates, etc.) than peak-to-trough ratios and therefore mathematical modeling is needed to understand the results. An interesting future direction of the field will be to include individual-based models that can evaluate the impact of bacterial clones with different distributions of growth/clearance rates, as well as working with experimental models (e.g., microfluidics) that would allow experimental investigation of the impact of single-cell level variation on total population behavior.

By combining mathematical modeling with direct quantification of bacterial population dynamics, we can gain insight into the major phenomena influencing colonization efficiency. Not only do these results help us understand the different steps of *B. theta* colonization, but they also serve as a proof of concept for studying other complex, multistep biological processes using the set of experimental and data-analysis tools we are describing here. This raises the possibility to optimize colonization conditions in order to promote the efficient engraftment of beneficial species into target microbiota or to better understand the processes of invasion of pathogens and the functional basis of colonization resistance.

# Methods

**Key resources table**

| Reagent type (species) or resource | Designation | Source or reference | Identifiers | Additional information |
|---|---|---|---|---|
| Genetic reagent (*Bacteroides thetaiotaomicron*) | *B. theta* WT | *Porter et al., 2017* | Not applicable | *tdk-*<br>Parent strain of *B. theta* VPI-5482 (ATCC 29148). Used to generate wild-type CPS mutants in this study. |
| Genetic reagent (*B. thetaiotaomicron*) | *B. theta* acapsular | *Porter et al., 2017* | Not applicable | *tdk- Δcps1-8*<br>Acapsular *B. theta* with deletion of capsular polysaccharide locus. Used to generate acapsular mutants in this study. |
| Genetic reagent (*B. thetaiotaomicron*) | *B. theta* WT barcode 1 | This study | PRJEB57876 (ERP142888) | *tdk-:: pNBU2-cat-ermG-GFP-WITS1*<br>*B. theta* WT strain isogenic barcode 1; GFP tag; erythromycin resistant. |
| Genetic reagent (*B. thetaiotaomicron*) | *B. theta* WT barcode 2 | This study | PRJEB57876 (ERP142888) | *tdk-:: pNBU2-cat-ermG-GFP-WITS2*<br>*B. theta* WT strain isogenic barcode 2; GFP tag; erythromycin resistant. |
| Genetic reagent (*B. thetaiotaomicron*) | *B. theta* WT barcode 11 | This study | PRJEB57876 (ERP142888) | *tdk-:: pNBU2-cat-ermG-GFP-WITS11*<br>*B. theta* WT strain isogenic barcode 11; GFP tag; erythromycin resistant. |
| Genetic reagent (*B. thetaiotaomicron*) | *B. theta* WT barcode 17 | This study | PRJEB57876 (ERP142888) | *tdk-:: pNBU2-cat-ermG-mCherry-WITS17*<br>*B. theta* WT strain isogenic barcode 17; mCherry tag; erythromycin resistant. |
| Genetic reagent (*B. thetaiotaomicron*) | *B. theta* WT barcode 19 | This study | PRJEB57876 (ERP142888) | *tdk-:: pNBU2-cat-ermG-mCherry-WITS19*<br>*B. theta* WT strain isogenic barcode 19; mCherry tag; erythromycin resistant. |

*Continued on next page*

*Continued*

| Reagent type (species) or resource | Designation | Source or reference | Identifiers | Additional information |
|---|---|---|---|---|
| Genetic reagent (*B. thetaiotaomicron*) | *B. theta* WT barcode 21 | This study | PRJEB57876 (ERP142888) | *tdk-:: pNBU2-cat-ermG-mCherry-WITS21*<br>*B. theta* WT strain isogenic barcode 21; mCherry tag; erythromycin resistant. |
| Genetic reagent (*B. thetaiotaomicron*) | *B. theta* acapsular barcode 1 | This study | PRJEB57876 (ERP142888) | *tdk- Δcps1-8:: pNBU2-cat-ermG-GFP-WITS1*<br>*B. theta* acapsular strain with isogenic barcode 1; GFP tag; erythromycin resistant. |
| Genetic reagent (*B. thetaiotaomicron*) | *B. theta* acapsular barcode 2 | This study | PRJEB57876 (ERP142888) | *tdk- Δcps1-8:: pNBU2-cat-ermG-GFP-WITS2*<br>*B. theta* acapsular strain with isogenic barcode 1; GFP tag; erythromycin resistant. |
| Genetic reagent (*B. thetaiotaomicron*) | *B. theta* acapsular barcode 11 | This study | PRJEB57876 (ERP142888) | *tdk- Δcps1-8:: pNBU2-cat-ermG-GFP-WITS11*<br>*B. theta* acapsular strain with isogenic barcode 1; GFP tag; erythromycin resistant. |
| Genetic reagent (*B. thetaiotaomicron*) | *B. theta* acapsular barcode 17 | This study | PRJEB57876 (ERP142888) | *tdk- Δcps1-8:: pNBU2-cat-ermG-mCherry-WITS17*<br>*B. theta* acapsular strain with isogenic barcode 17; mCherry tag; erythromycin resistant. |
| Genetic reagent (*B. thetaiotaomicron*) | *B. theta* acapsular barcode 19 | This study | PRJEB57876 (ERP142888) | *tdk- Δcps1-8:: pNBU2-cat-ermG-mCherry-WITS19*<br>*B. theta* acapsular strain with isogenic barcode 19; mCherry tag; erythromycin resistant. |
| Genetic reagent (*B. thetaiotaomicron*) | *B. theta* acapsular barcode 21 | This study | PRJEB57876 (ERP142888) | *tdk- Δcps1-8:: pNBU2-cat-ermG-mCherry-WITS21*<br>*B. theta* acapsular strain with isogenic barcode 21; mCherry tag; erythromycin resistant. |
| Genetic reagent (*B. thetaiotaomicron*) | *B. theta* WT untagged | This study | PRJEB57876 (ERP142888) | *tdk-:: pNBU2-bla-tetQb*<br>Untagged strain. *B. theta* WT strain with GFP insert; tetracycline resistant. |
| Genetic reagent (*B. thetaiotaomicron*) | *B. theta* acapsular untagged | This study | PRJEB57876 (ERP142888) | *tdk- Δcps1-8:: pNBU2-bla-tetQb*<br>Untagged strain. *B. theta* acapsular strain with GFP insert; tetracycline resistant. |
| Genetic reagent (*Salmonella enterica*) | Stm attenuated (M3103) | *Diard et al., 2017* | Not applicable | *ΔssaV*<br>*Salmonella enterica* serovar Typhimurium (SL1344), attenuated (SPI-2 KO) |
| Genetic reagent (*S. enterica*) | Stm avirulent (M2702) | *Diard et al., 2017* | Not applicable | *ΔinvG ΔssaV*<br>*Salmonella enterica* serovar Typhimurium (SL1344), avirulent (SPI-1 KO and SPI-2 KO). |
| Sequence-based reagent | WITS01_F | *Maier et al., 2014* | Forward primer barcoded strain 1 | acgacaccactccacaccta |
| Sequence-based reagent | WITS02_F | *Maier et al., 2014* | Forward primer barcoded strain 2 | acccgcaataccaacaactc |
| Sequence-based reagent | WITS11_F | *Maier et al., 2014* | Forward primer barcoded strain 11 | atcccacacactcgatctca |
| Sequence-based reagent | WITS17_F | *Maier et al., 2014* | Forward primer barcoded strain 17 | tcaccagcccaccccctca |
| Sequence-based reagent | WITS19_F | *Maier et al., 2014* | Forward primer barcoded strain 19 | gcactatccagccccataac |
| Sequence-based reagent | WITS21_F | *Maier et al., 2014* | Forward primer barcoded strain 21 | acaaccaccgatcactctcc |
| Sequence-based reagent | ydgA_R | *Maier et al., 2014* | Common reverse primer for all tagged strain | ggctgtccgcaatgggtc |
| Sequence-based reagent | BTt70-CHF | *Jacobson, 2017* | pNBU2 vector genome integration test | TTCAAATTGCTCGGTAAAGCTC |

*Continued on next page*

*Continued*

| Reagent type (species) or resource | Designation | Source or reference | Identifiers | Additional information |
|---|---|---|---|---|
| Sequence-based reagent | BTt70-CHR | *Jacobson, 2017* | pNBU2 vector genome integration test | AAAACCTTGATTTTACGGGAC |
| Sequence-based reagent | BTt71-CHF3 | *Jacobson, 2017* | pNBU2 vector genome integration test | TTCGAGGAATGAAGCATCTCCGTA |
| Sequence-based reagent | BTt71-CHR3 | *Jacobson, 2017* | pNBU2 vector genome integration test | ACCGTTCCGATTCAATTTCGT |
| Sequence-based reagent | IntN2BTt71-CHF3 | *Jacobson, 2017* | pNBU2 vector genome integration test | TTTCCGGCTCTCCAATGCAA |

## Bacterial strains and cultures

*B. theta* strains were grown anaerobically (5% $H_2$, 10% $CO_2$, rest $N_2$) at 37°C, overnight in brain heart infusion (BHI) supplemented media (BHIS: 37 g/L BHI [Sigma]; 1 g/L-cysteine [Sigma]; 1 mg/L Hemin [Sigma]). For enrichment cultures in plates, we used BHI-blood agar plates (37 g/L BHI [Sigma]; 1 g/L--cysteine [Sigma]; 10% v/v defribinated sheep blood [Sigma]). Antibiotics were added to liquid cultures or plates as required for strain selection: erythromycin 25 µg/L or tetracycline 2 µg/L. In the case of BHI-blood agar plates used for cloning or gut content enrichment, we additionally added gentamycin 200 µg/L to all plates to prevent growth of other microbiota species. Plates were incubated for 48–72 hr at 37°C in anaerobic conditions. For a complete list of the bacterial strains used in this study, see Key Resources Table.

## Isogenic barcode construction and integration

Genetic tags, fluorescent proteins, and antibiotic resistance were introduced by using the mobilizable *Bacteroides* element NBU2, which integrates into the *Bacteroides* genomes at a conserved location at either BTt70 or BTt71 (*Wang et al., 2000*). Gene fragments containing a unique 40 bp sequence (biding site for forward primer) and a 609 bp sequence with the ydgA pseudogene (common binding site for reverse primer) were synthesized (gBlocks, Integrated DNA Technologies) and cloned by Gibson Assembly Master Mix (NEB) into an NBU2 plasmid carrying the erythromycin-resistant cassette ermG (barcoded *B. theta* strains) and a fluorescent GFP or mCherry protein (see Key Resources Table for specific combination of fluorescent protein and tag). A similar NBU2 plasmid carrying the tetracycline-resistant cassette tetQb and the GFP protein was used to construct the untagged strains (*B. theta* untagged). All fluorescent protein genes have high-expression promoter and an optimized RBS (*Whitaker et al., 2017*). For both, we used 10 µL of desalted assembly reaction products to transform competent *E. coli* S17-1 cells (mid-log cells, washed three times in deionized ice-cold water) by electroporation (V = 1.8 kV; MicroPulser, Bio-Rad). After 1 hr recovery at 37°C in 1 mL of LB, cells were plated on LB plates with chloramphenicol (12 µg/mL) and grown overnight. Plasmid-carrying *E. coli* S17-1 and *B. theta* strains were cultured overnight in 5 mL of liquid media. *E. coli* S17-1 and *B. theta* were washed with PBS, pooled in 1 mL of PBS, plated BHI-blood agar plates without antibiotics, and grown aerobically at 37°C for at least 16 hr. The lawn of *E. coli* S17-1 and *B. theta* was collected in 5 mL of PBS, homogenized by vortex, and 100 µL were plated in BHI-blood agar plates supplemented with erythromycin 25 µg/L and gentamycin 200 µg/L. After 48 hr, single colonies were streaked in fresh BHI-blood agar plates with antibiotics to avoid potential contamination with WT strains. Successful insertion in the BTt70 or BTt71 sites was evaluated by PCR (Key Resources Table). To minimize potential variation, we used strains with a single insertion in BTt70. In summary, the barcoded strains carried the barcode, erythromycin resistance cassette and fluorescent protein (GFP or mCherry) in the genome. The untagged *B. theta* carried a tetracycline resistance cassette together with a GFP protein inserted at the same position in the genome.

## Quantification of barcoded *B. theta* from mixed samples

One investigator blinded to sample metadata (i.e., microbiota, bacteria strain used, acute challenge used) performed the sample processing and qPCR quantification of barcodes. Samples were serially diluted and plated on appropriate selective BHIS agar (gentamycin 200 µg/L plus either erythromycin 25 µg/L or tetracycline 2 µg/L) and cultured in 5% $H_2$, 10% $CO_2$, rest $N_2$, at 37°C for 48 hr (Coy Anaerobic tent). CFU determination was carried out by counting, then colonies were washed from the plates (all plates with at least 30 colonies were included), pooled in 5 mL of PBS and homogenized by vortex. Genomic DNA was isolated with the QIAamp DNA Mini Kit (QIAGEN). qPCR was performed using with FastStart Universal SYBR Green Master Mix (Roche, Cat# 4385610). Primers (Key Resources Table) were mixed to a final concentration of 1 µM. Between 160 and 200 ng of DNA was amplified in duplicates using StepOne Plus or QuantStudio 7 Flex instruments (Applied Biosystems) using the following protocol: initial denaturation at 95°C for 14 min followed by 40 cycles of 94°C for 15 s, 61°C for 30 s, and 72°C for 20 s as described previously. As qPCR reactions for these barcodes have identical efficiencies, genomic DNA extracted from a single barcoded strain was used as an internal standard and CT values were backcalculated to this standard curve to generate a relative frequency of each barcode in the pooled colonies. These relative frequencies were then multiplied by the total CFU/g of barcoded *B. theta* to obtain the CFU/g of each barcoded strain.

## In vitro growth curves and competition

Individual *B. theta* strains were grown overnight on BHIS. Stationary-phase cultures were washed with PBS, and O.D. was quantified and adjusted to 0.05 in 200 µL of fresh BHIS in a round 96-well tissue culture plates. Plates were transferred into the anaerobic tent and growth was quantified at 37°C with shaking using a plate reader (Infinite PRO 200, Tecan).

For competition experiments, stationary-phase cultures *B. theta* WT or acapsular strains were washed with PBS, O.D. was quantified and adjusted to approximately 5 × 106 CFU/mL per strain (one *B. theta* untagged and six *B. theta* barcoded strains) in 10 mL of fresh BHIS. An aliquot of this mix was serially diluted and plated in BHI-blood agar plates with the respective antibiotics for CFU quantification. Cultures were kept overnight, with shaking (800 rpm) at 37°C in the anaerobic tent. Afterward, an aliquot was plated as described before. For assessing the competition between *B. theta* barcoded strains, we isolated DNA from one of the dilutions used for quantification and assessed the relative distribution of the tags by qPCR (see 'Quantitative PCR' section). For assessing the competition among strains with different antibiotic resistances, we calculated the competition index by dividing the CFU/mL of the untagged *B. theta* untagged strain (tetracycline-resistant) by the adjusted number of *B. theta* barcoded strains (erythromycin-resistant; CFU/mL divided by six, as all the barcoded strains were present in the culture).

## Mice

All animal experiments were performed with approval from the Zürich Cantonal Authority under license number ZH120/19 and ZH009/21. In all experiments, we used mice with C57BL/6J genetic background, between 12 and 15 weeks old and of variable gender. C57BL/6J GF and gnotobiotic mouse lines (LCM [*Stecher et al., 2010*]; OligoMM12 [*Brugiroux et al., 2016*]) were raised in surgical isolators under high hygiene standards at the ETH Phenomic Center and were regularly tested for contamination by aerobic and anaerobic cultivation, culture-independent assessment of intestinal bacterial densities and serology/PCR for common viruses and eukaryotic pathogens. Note that all LCM mouse lines were bred for 1 year (OligoMM12) or more than 10 years (LCM) with their gnotobiotic microbiota. C57BL/6J SPF mouse line was raised in IVC cages in a different barrier unit of the same facility. Mice were transferred to our experimental facility in sterile, tight closed cages and house into the IsoCage P- Bioexclusion System (Tecniplast) for 24–48 hr before the experiment to adapt to new housing conditions. In all experiments, standard chow and water was prepared under strict aseptic conditions to avoid any potential contaminations. Although mice themselves were not randomized on each experiment, cages containing appropriate mouse numbers were randomly assigned to each inoculum/treatment and both genders are represented in all groups.

## In vivo growth curves and competition

*B. theta* WT WITS 01 or *B. theta* acapsular WITS 01 strain was grown overnight in BHIS. Stationary-phase cultures were washed with PBS, and an inoculum of ~5 × 10$^7$ CFUs/100 μL dose was prepared. C57BL/6J mice carrying the described microbiota composition (LCM, OligoMM12, SPF, see figure legends for specific group numbers) were gavaged with the inoculum (either *B. theta* WT or *B. theta* acapsular). Fecal pellets were collected approximately every 4 hr post-inoculation. Fecal pellets were weighted and homogenized in 500 μL of PBS with steel ball by mixing (25 Hz, 2.5 min) in a TissueLyser (QIAGEN). Serial dilutions were plated for quantification on BHI-blood agar plates supplemented with gentamycin 200 μg/L and erythromycin 25 μg/L and cultured in anaerobic conditions. Note that zero colonies grew from the feces of LCM, OligoMM12, and SPF mice on BHIS plates with gentamycin and erythromycin.

For competition experiments, *B. theta* strains were grown overnight in 8 mL of BHIS with corresponding antibiotics (erythromycin 25 μg/L or tetracycline 2 μg/L). Each culture was spun down at 3000 × *g* for 20 min and resuspended in 10 mL of PBS and individual O.D. was measured (cell number estimation 1 O.D. = ~4 × 10$^8$ cells/mL). Each strain was adjusted to approximately 5 × 10$^6$ CFU/100 μL dose per strain in the inoculum mix (one *B. theta* untagged and six *B. theta* barcoded strains). GF mice were gavaged with 100 μL of the inoculum. After 48 hr, fecal and cecal contents were collected. Fecal content was homogenized and plated for quantification as described before. Cecal content was resuspended in 1 mL of PBS and homogenized with steel ball by mixing with the same protocol (25 Hz, 2.5 min). Serial dilutions were prepared and plated in BHI-blood agar plates supplemented with gentamycin plus either erythromycin or tetracycline for CFU quantification of each strain. Like the in vitro competition experiment, we isolated DNA from one of the dilutions used for quantification to assess the competition between *B. theta* barcoded strains. Relative distribution of the tags was obtained by qPCR. For calculating the competition index among strains with different antibiotic resistances, we divided the bacteria density of the untagged *B. theta* untagged strain by a sixth of the bacteria density of the total *B. theta* barcoded strains (as all six barcoded strains were present in the culture).

## Colonization experiments

Stationary-phase *B. theta* cultures were prepared overnight as described before. Each culture was washed with PBS to remove residual antibiotics and adjusted in the inoculum based on its O.D.600nm. Unless otherwise stated, the untagged strain (*B. theta* untagged) was present at ~5 × 10$^7$ CFUs/100 μL dose of the inoculum. For the *B. theta* barcoded strains, we prepared an initial 1:1:1:1:1:1 mix of all six barcoded strains in 50 mL of PBS at a concentration of 10$^5$ CFU/mL of each strain. After mixing by vortex for 1 min, the required amount of *B. theta* barcoded strains was prepared by serial dilution and spiked into the inoculum (between 30 and 5 × 10$^4$ CFUs depending on the experiment). LCM (Oligo) and SPF C57BL/6J mice were gavaged with the 100 μL inoculum. To check composition, the inoculum was serially diluted and plated for quantification of CFUs in BHI-blood agar plates supplemented with gentamycin 200 μg/L plus either erythromycin 25 μg/L or tetracycline 2 μg/L. In addition, three whole doses (100 μL) were directly plated on three BHI-blood agar plates with gentamycin plus erythromycin or tetracycline (as appropriate) to address initial distribution of *B. theta* barcoded strains in the inoculum by quantitative PCR (qPCR). Unless otherwise stated, 2 days after colonization, mice were euthanized and cecal content was collected in 2 mL Eppendorf tubes and weighted. Cecal content was homogenized as described before. Serial dilutions were prepared and plated in BHI-blood agar plates supplemented with gentamycin plus either erythromycin or tetracycline for CFU quantification. In addition, 100 μL of homogenized content was plated directly in BHI-blood agar plates with gentamycin plus erythromycin to generate biomass of the assessment of the distribution of *B. theta* barcoded strains by qPCR.

## In vivo competition of post-colonization versus original strains

To discard potential increased colonization fitness in the barcoded strains that were present in the cecum content after 48 hr, we isolated single *B. theta* WT barcoded strains that were present in the cecal content of SPF mice during a colonization experiment. Single colonies were expanded in liquid media, and the presence of a single strain was confirmed by qPCR analysis of the barcodes present. We randomly selected three of the *B. theta* WT barcoded strains isolated from the cecal content.

We prepared an inoculum as described before for the in vivo competition experiments with approximately $5 \times 10^6$ CFU/100 µL of each strain in the inoculum mix. We complemented the inoculum with the remaining three *B. theta* WT barcoded strains coming from the original stock. SPF mice were inoculated by gavage and cecal content was collected 48 hr later. Cecal content was processed as described before for CFU quantification and relative barcode distribution by qPCR.

## Diet modification and infection challenge experiments

In accordance with what we described before, *B. theta* WT strains were grown overnight in BHIS with corresponding antibiotics. As the untagged strain *B. theta* untagged was used in higher concentrations, we prepared between 50 and 100 mL of liquid culture depending on the number of mice to colonize. Inoculum was prepared as previously described with a concentration of $10^8$–$10^9$ CFUs/100 µL dose of untagged *B. theta* untagged, spiked with approximately 30 CFU of each *B. theta* barcoded strains. GF mice were gavaged with 100 µL of the inoculum. Mice were maintained on standard chow diet (Kliba Nafag, 3537; autoclaved; per weight: 4.5% fat, 18.5% protein, ~50% carbohydrates, 4.5% fiber) for 4 days. Afterward, mice were housed on fresh IsoCages, and challenges were applied as follows: (1) control group (continuation of standard chow diet); (2) Western-type diet without fiber (BioServ, S3282; 60% kcal fat; irradiated; per weight: 36% fat, 20.5% protein, 35.7% carbohydrates, 0% fiber); or (3) infection with $5 \times 10^7$ CFU of attenuated *Salmonella* Typhimurium (Stm SL1344$^{\Delta SPI-2}$). Fecal pellets were collected pre-challenge (day 0) and during the following 3 days. On day 3, mice were euthanized and cecal content was collected. Fecal pellets were weighted and homogenized in 500 µL of PBS as described before. Serial dilutions were prepared and plated in BHI-blood agar plates supplemented with corresponding antibiotics for CFU quantification. In addition, 100–300 µL of homogenized content was plated directly for further assessment of the distribution of *B. theta* barcoded strains by qPCR. Cecal content was processed as previously described.

## DNA extraction for community composition analysis and growth estimates

To assess microbial community composition, fecal pellets from LCM mice and cecum content from SPF mice were obtained and flash frozen. To generate growth estimates of *B. theta* in an OligoMM12 background, both flash-frozen fecal pellets and cecum content were used. For enzymatic lysis, half a fecal pellet or roughly 30 mg of flash-frozen cecum content per sample were incubated in 100 µL of 1× TE buffer (30 mM Tris-HCl and 1 mM EDTA) supplemented with 30 mg/mL Lysozyme (Sigma-Aldrich), 1.6 U/mL Proteinase K (New England Biolabs), 10 U/mL Mutanolysin (Sigma-Aldrich), and 1 U/µL SUPERase•In RNase Inhibitor (Invitrogen) at room temperature for 10 min. To aid disruption, one 2 mm metal bead was added, and the samples were vortexed every 2 min. Subsequently, the samples were mixed with 550 µL RLT Plus buffer (QIAGEN) complemented with 5.5 µL 2-beta-mercaptoethanol (Sigma-Aldrich) and prefilled tubes with 100 µm Zirconium beads (OPS Diagnostics LLC). The samples were disrupted twice at 30 Hz for 3 min using the mixer mill Retsch MM400 with 5 min incubation at room temperature between each disruption. DNA was extracted from all samples with the DNA/RNA Mini kit (QIAGEN) following the standard protocol and eluting the DNA in 100 µL elution buffer (EB). For the LCM samples, three negative extraction controls and three negative PCR controls were included. For the SPF samples, one water sample was used as negative extraction control and subsequently split into three negative library controls undergoing the same library preparation as all samples. The integrity and quality of the extracted DNA was assessed on a Qubit and Fragment Analyzer, respectively. The DNA was purified by overnight ethanol precipitation at −20°C in 275 µL ice-cold Ethanol (Sigma-Aldrich), 10 µL 3 M sodium acetate (Invitrogen), and 1 µL 20 mg/mL glycogen (Invitrogen) with subsequent centrifugation at 4°C for 30 min and two wash steps in 500 µL ice-cold 75% ethanol with centrifugation at 4°C for 10 min each time. The DNA purity was assessed on a Nanodrop.

## 16S sequencing for LCM, OligoMM12, and SPF community composition analysis

16S amplicon libraries were generated from 50 ng input DNA with the Illumina primer set 515F (*Parada et al., 2016*) and 806R (*Apprill et al., 2015*), 12 cycles in PCR 1 and 13 cycles in PCR 2. Three positive controls containing 11 ng input DNA of ZymoBIOMICS Microbial Community DNA Standard II (Zymo

Research) were used. Illumina Unique Dual Indexing Primers (UDP) were used for library multiplexing. A 12 pM library pool spiked with 20% PhiX was sequenced at the Functional Genomics Center Zurich (FGCZ) using the MiSeq platform and 2 × 300 bp PE-reads with a target fragment size of 450 bp, resulting in approximately 60,000 and 400,000 reads per sample for the LCM and OligoMM12/SPF sequencing runs, respectively. Raw sequencing data from LCM and SPF mice can be accessed on ENA (https://www.ebi.ac.uk/ena/browser/home) under Project ID PRJEB57876. The OligoMM12 was previously published and can be accessed on ENA under the Project ID PRJEB53981 (*Hoces et al., 2022*).

## Metagenomic sequencing for *B. theta* growth estimates in OligoMM12 background

Genomic DNA was sheared to a target fragment size of 350 bp length with the ultrasonicator Covaris LE220 following a standard protocol (30 µL volume, 220 W peak incident power, 89 s treatment time). Metagenomic libraries were prepared from 10 ng sheared DNA with the NebNext Ultra II DNA Library Prep Kit for Illumina. Sample-specific adaptations included tenfold adapter dilution, no size selection by adding 1 volume (89 µL) of Cytiva Sera-Mag Select beads in the first cleanup and eight PCR cycles in the amplification step. Nebnext Multiplex Oligos for Illumina (Dual Index Primer Set 1) were used for library multiplexing. The final cleanup was done with a left side size selection by adding 0.7 volumes (35 µL) of Cytiva Sera-Mag Select beads. A 1 nM library pool spiked with 3% PhiX was sequenced at the FGCZ using the NextSeq2000 platform and 2 × 150 bp PE-reads with a target fragment size of 500 bp, resulting in approximately 30,000,000 reads per sample.

## Data analysis

### 16S community composition analysis

Raw sequencing reads from all samples and 3–6 positive/negative controls served as input for the inference of ASVs using dada2 v1.22 (*Callahan et al., 2016*). Primer sequences (515F, 806R) were removed using cutadapt v2.8 (*Martin, 2011*), and only inserts that contained both primers and were at least 75 bases were kept for downstream analysis. Next, reads were quality filtered using the filterAndTrim function of the dada2 R package (maxEE = 2, truncQ = 3, trimRight = (40, 40)). The learnErrors and dada functions were used to calculate sample inference using pool = pseudo as parameter. Reads were merged using the mergePairs function and bimeras were removed with the removeBimeraD-enovo (method = pooled). Remaining ASVs were then taxonomically annotated using the IDTAXA classifier (*Murali et al., 2018*) in combination with the Silva v138 database (*Quast et al., 2013*) available at http://www2.decipher.codes/Downloads.html. The resulting ASV table was used to check for contaminations with the decontam R package (*Davis et al., 2018*) using both frequency-based and prevalence-based classification with a single probability threshold of 0.05 computed by combining both probabilities with Fisher's method (method = combined). ASVs classified as contaminants as well as the positive/negative controls were excluded from downstream analyses. The remaining ASV abundance table was downsampled to a common sequencing depth (28,000 reads per sample for LCM and 190,000 reads per sample for Oligo/SPF) to correct for differences in sequencing depth between samples using the rrarefy function of the vegan R package. Relative abundance plots at different taxonomic levels were generated (LCM at species level, OligoMM12 at strain level, SPF at family level).

For assessing the LCM composition, ASVs were clustered at 97% sequence identity with VSEARCH (usearch_global) (*Rognes et al., 2016*), which resulted in eight distinct ASVs with a maximum sequence identity of 96% between the two most similar ASVs. These ASVs were annotated at species level by alignment to 16S sequences of known community members from the original Schaedler flora (ASV01), from the OMM12 community (ASV03, ASV04, ASV08) and by alignment against the Silva v138 database. Due to annotation inconsistencies, ASV05 could only be annotated at the family level. The OligoMM12 strains were identified using the package bio for rRNA sequence extraction from the GenBank accessions described earlier (*Hoces et al., 2022*) and the tool VSEARCH (search_exact) (*Rognes et al., 2016*) for sequence alignment to the 16S sequences from the detected ASVs. ASVs with a mean relative abundance below 0.05% across all samples were grouped into 'Other.' *Megasphaera* was detected at a mean relative abundance of 0.06% but was also grouped into the category 'Other' since it was not knowingly part of the original OligoMM12 community. The category 'Other' in total amounted to roughly 0.11% of the total relative abundances, thus the oligo strains represented at least 99.8% of the detected ASV abundances. For the SPF community composition, ASVs were

clustered at family level. ASVs with mean relative abundance below 1% or without taxonomic annotation at family level were grouped into the category 'Other.' The category 'Other' in total amounted to roughly 5% of the total relative abundances.

## Metagenomic analysis for growth rate estimation

Sequencing reads from all metagenomic samples of *B. theta* in OligoMM12 background from feces and cecum were quality filtered using BBMap (v.38.71; *Bushnell, 2014*) by removing sequencing adapters from the reads, removing reads that mapped to quality control sequences (PhiX genome) and discarding low-quality reads using the parameters trimq = 14, maq = 20, maxns = 0, and minlength = 45. The in situ growth rate prediction tool CoPTR (*Joseph et al., 2022*) was used to compute growth rate estimates from the quality-controlled metagenomic reads by aligning them against a database containing all 12 OligoMM12 genomes available under Bioproject PRJNA317592 and the *B. theta* genome available under GenBank accession number CP092641.1.

## Identification of genomic variants among *B. theta* strains

Genomic DNA from all *B. theta* strains was isolated from overnight cultures using the QIAamp DNA Mini Kit (QIAGEN). Samples were sent for whole genome sequencing at Novogene. Data preprocessing pipeline for adapter trimming and contaminant filtering is described at https://methods-in-microbiomics.readthedocs.io/en/latest/preprocessing/preprocessing.html. The raw reads for each strain were trimmed and filtered using BBMap v. 38.18 (*Bushnell, 2014*). The reads were mapped against *Bacteroides thetaiotaomicron* strain VPI 5482 genome (CP092641.1) using bwa v. 0.7.17 (*Li and Durbin, 2009*). The resulting bam files were sorted according to the coordinates and indexed using samtools-1.9 (*Danecek et al., 2021*). The duplicated reads were removed using gatk v.4.2 MarkDuplicates (*McKenna et al., 2010*). Variant calling and filtering (bcftools filter -Ov -sLowQual -g5 -G10 -e 'QUAL <10 || DP4[2]<10 || DP4[3]<10 ||(DP4[2]+DP4[3])/sum(DP4)<0.9 || MQ <50') was performed using Bcftools v1.133. The variant annotation was done using snpEff (*Cingolani et al., 2012*).

## Mathematical modeling overview (see Appendix 1 'Supplementary methods' for more detailed description)

### Estimation of colonization probability based on lost tags

Let us denote $C$ the bacterial concentration in the prepared solution. If we have volume $V$ of this solution, then there are $N = CV$ bacteria. Therefore, the probability to have taken $n_0$ starting bacteria into an inoculum of volume $v_0$ is

$$p\left(n_0\right) = Binomial\ distribution\left(N, \frac{n_0}{V}\right) = \left(\frac{v_0}{V}\right)^n \left(1 - \frac{v_0}{V}\right)^{N-n_0} \frac{N!}{\left(N - n_0\right)!n_0!} \tag{1}$$

when $N = cV$ is large and $v_0 \ll V$,

$$p\left(n_0\right) \cong Poisson\ distribution\left(N\frac{v_0}{V}\right) = \frac{\left(N\frac{v_0}{V}\right)^{n_0} expexp\left(-N\frac{v_0}{V}\right)}{n_0!} \tag{2}$$

We define $\beta$ as the probability for each bacterium to get to the cecum alive, and then have its lineage survive until measurement. Logically, the probability for a barcoded *B. theta* strain **not** to be present at measurement time is the zero of the Poisson distribution of average $\beta n_0$, and thus

$$p_{loss} = expexp\left(-\beta n_0\right) \tag{3}$$

$n_0$ is estimated via the concentration and volume of the inoculum, and $p_{loss}$ is estimated via the number of tags lost divided by the total number of tags. Therefore, $\beta$ is estimated as

$$\beta \cong \frac{-loglog\left(\frac{n_{loss\ tags}}{n_{tags}}\right)}{n_0} \tag{4}$$

To consider the fact that not all tags have the same $n_0$ when we pool data from multiple experiments, $\beta$ is actually estimated by maximizing the probability of the experimental observations:

$$LLL = \sum_{i=1}^{\omega} \log \left( \left( \exp \left( -\beta n_i \right) \right)^{l_i} \left( 1 - \exp \left( -\beta n_i \right) \right)^{1-l_i} \right) \tag{5}$$

This expression is also used for calculating the confidence interval, as detailed in Appendix 1.

## Estimation of colonization probability based on variance

The variance on the proportions is

$$var \left( p \right) = \frac{1}{h_1} \sum \left( p_i - \frac{1}{h} \right)^2 = \frac{1}{h_1} \sum \left( \frac{n_i}{\sum n_j} - \frac{1}{h} \right)^2 \tag{6}$$

In the limit where the initial number of bacteria are of the same order of magnitude, we find

$$\langle var \left( p \right) \rangle - var \left( p_0 \right) \cong \frac{1}{h \sum n_{j,0}} \frac{var1}{m1^2} \tag{7}$$

with $var \left( p_0 \right)$ the variance in proportions in the inoculum, $\sum n_{j,0}$ the total number of tags in the inoculum, and $var1/m1^2$ the relative variance starting from one bacterium. We find (see Appendix 1 'Supplementary methods') that $var1/m1^2$ is 2/(colonization probability). $var1/m1^2$ can be estimated for each mouse using **equation (7)**, and the average variance is used to estimate $var1/m1^2$. The standard error on $var1/m1^2$ is used to obtain the confidence interval for the colonization probability.

## Estimation of clearance rate due to flow

We examined the expected magnitude of the effect of an extended lag phase in the cecum on colonization probability to determine whether this is consistent with our observed neutral tagging data. It should be noted that the cecum is a dynamic environment with pulsatile arrival of material from the small intestine and loss of material to the feces. This generates a clearance rate due to flow on top of any clearance rate due to bacterial death. Assuming that the main site of growth of *B. theta* is the cecum/upper colon, the parameter for clearance due to flow can be estimated by quantifying the volume of cecum content lost per day. This can be empirically estimated by measuring (1) fecal dry mass produced per day, and (2) the water content of cecum content. Assuming minimal change in dry mass during colon transit in the mice, this infers a dilution rate of cecal content in the order of 0.12 volumes/hr in a GF mouse and 0.18 volumes/hr in an SPF mouse; LCM mice will have a value in between these two. Bacterial clearance due to killing will contribute over and above these values. Of note, bacteria with a long lag phase after introduction into the cecum will be cleared by the flow before growth can start, that is, during the early phase of colonization this will be a determinant of colonization probability.

## Estimation of cecum turnover rates

| | Water fraction in cecal mass (%) | Dry fecal excretion (g/day) | Estimated wet cecal mass excretion (g/day) | Wet cecal mass (g) | Estimated cecum turnover rate (volume/day) | Estimated cecum turnover rate (volume/hr) |
|---|---|---|---|---|---|---|
| Germ-free | 80.9 (0.4) | 1.55 (0.27) | 8.12 (1.42) | 2.83 (0.59) | 2.87 (0.78) | 0.12 (0.03) |
| Specific pathogen free | 76.2 (1.2) | 0.81 (0.09) | 3.40 (0.42) | 0.77 (0.32) | 4.42 (1.91) | 0.18 (0.08) |

## Estimation of the competitive index

We assume that bacteria have first a probability of survival $q_i$ (with $i = w$ for the WT strain, and $i = a$ for the acapsular strain). Then once the cecum is reached, they have a loss rate $c_i$.

During an initial lag-phase $\tau_i$, *B. theta* does not grow. On exit from lag phase, each bacterial strain grows logistically, initially at a rate $r_i$, which saturates when approaching carrying capacity $K$ with a factor $\left( 1 - \left( A + W \right) /K \right)$. $A$ and $W$ denote the population density of acapsular and wildtype *B. theta*, respectively, when carrying capacity is reached, the total number of bacteria remains constant until the end of the experiment at time $t_{tot}$, with both loss and replication ongoing and compensating each

other. Given the growth rates for WT and acapsular *B. theta* are similar in vitro, we assume $r_w = r_a$ also in vivo. Correspondingly, the difference in the initial net growth rates $(r_i - c_i)$ considered to originate from differences in the in vivo clearance rates $c_a > c_w$. In the competition setting at carrying capacity, the majority of the population is composed of WT *B. theta*, such that the global population size is cleared with rate approximately $c_w$. By definition, to maintain the total population size the effective replication rate of both strains of *B. theta* must exactly compensate this loss rate, $c_w$. However, as the acapsular strain implicitly has a higher clearance rate $c_a$, their net growth rate at carrying capacity becomes negative, that is, although the total population size remains constant, the acapsular *B. theta* population size will continuously decrease in frequency over time.

With this model (see detailed calculations in Appendix 1 'Supplementary methods'), we find that the relative ratio between WT and acapsular is

$$\frac{q_\omega}{q_a} exp\ exp\ ((net_w + c_\omega)(\tau_a - \tau_w))(exp(net_w - net_a)t_{tot}) \tag{8}$$

For all the microbiota except SPF, the colonization probabilities ($q$) were similar for the WT and acapsular, as determined for single colonizations of LCM and OligoMM12 mice. Therefore, we assume $q_a = q_w$. For SPF, we use the ratio of $q_w/q_a$ from the single-colonization experiments, adjusted for the fact that the full colonization probability also includes steps after the initial death before reaching the cecum. All the parameters used were determined from single-colonization experiments in the relevant microbiota backgrounds.

## Estimation of colonization probability during competition

The overall survival probability for acapsular *B. theta* in the competition experiment is the colonization probability from single-colonization experiments, multiplied by a factor considering later loss (when the carrying capacity is reached by the WT and the acapsular decreases). The complete expression can be found in the corresponding section of Appendix 1 'Supplementary methods'.

## Estimation of survival probability after challenge

In these experiments, at the time of the start of the challenge, the bacterial population is at carrying capacity, so the net growth rate is zero (i.e., the growth rate [likely limited by availability of nutrients] is the same as the loss rate due to flow/clearance). We also assume that the population size is known at the start of the challenge. The challenge may have different effects: it may impose a temporary bottleneck in the population (loss becomes higher than reproduction) or it may increase the loss rate (with the reproduction rate increasing enough to compensate), and thus the turnover of the population. In any case, we can calculate $\beta$ as the probability that a bacteria present at day = 0 of the challenge has its lineage still alive and detectable in cecum content at day 3 via either mechanism. If there are $n_0$ bacteria carrying a given barcode at day = 0, then

$$p_{loss} = \left(1 - \beta\right)^{n_0} \tag{9}$$

To estimate the total population size in the cecum before the challenge ($n_0$), we assume that (1) all the animals are colonized at steady state at day = 0, and (2) the cecum mass and bacteria concentration is the same on day = 0 in all mice as it is on day = 3 in the control group. As there are small but relatively consistent differences in total CFU between feces and cecum, we used the average relationship between feces and cecum CFU in control mice on day = 3 to estimate the bacterial concentration in the cecum at day = 0. We assumed cecum barcoded strain abundance based on qPCR/plating and concentration correction of fecal samples on day = 0. In addition, $p_{loss}$ was estimated via the number of barcodes lost from cecum content on day = 3 divided by the total number of barcoded strains across all mice analyzed. $\beta$ can then be estimated based on *equation (6)*.

## Statistical analysis

Sample size was determined based on previous experiments (*Maier et al., 2014*; *Porter et al., 2017*; *Wotzka et al., 2019*) using at least five mice per group where large effect size was expected. All group sizes are described in the figure legends.

Where errors are expected to be log-normal distributed (e.g., CFU density comparisons), all statistical tests were carried out on log-normalized data. One-way ANOVA followed by Tukey's honest significance test was used for comparison of three or more groups. For model-inferred parameters,

we compared mean and standard deviation calculated as described in the previous section. No data points were omitted from statistical analysis or for the estimation of parameters. Statistical analysis was performed with RStudio v1.2 and R v3.6.

## Resource availability

### Lead contact

Any further communication, including those related to resource sharing, may be directed to, and fulfilled by, the lead contact Emma Slack (emma.slack@hest.ethz.ch).

### Materials availability

All strain and material generated in this study are available upon request to the corresponding author.

## Acknowledgements

We express our gratitude to Sven Nowok and Dominik Bacovcin for their great support in maintaining the gnotobiotic mouse facility and to the whole team in EPIC and RCHCI. Also, we thank Dr. Annika Hausmann and Verena Lentsch for the discussions about this project. We acknowledge Prof. Bärbel Stecher for provision of the OligoMM12 bacterial strains used to generate the mouse colony used in these experiments.

This work was funded by NCCR Microbiomes, a research consortium financed by the Swiss National Science Foundation (ES, SS, W-DH); Swiss National Science Foundation (40B2-0_180953, 310030_185128) (ES), European Research Council Consolidator Grant (no. 865730-SNUGly) (ES), Gebert Rüf Microbials (GR073_17) (ES, CL); Botnar Research Centre for Child Health Multi-Invesitigator Project 2020 (BRCCH_MIP: Microbiota Engineering for Child Health) (ES, SS), Agence Nationale de la Recherche (ANR-21-CE45-0015, ANR-20-CE30-0001) (CL), and MITI CNRS AAP adaptation du vivant à son environnement (CL). The funders had no role in study design, data collection and analysis, decision to publish, or preparation of the manuscript.

---

## Additional information

### Funding

| Funder | Grant reference number | Author |
| --- | --- | --- |
| Schweizerischer Nationalfonds zur Förderung der Wissenschaftlichen Forschung | NCCR Microbiome | Wolf-Dietrich Hardt Shinichi Sunagawa Emma Slack |
| Gebert Rüf Stiftung | GR073_17 | Claude Loverdo Emma Slack |
| Botnar Research Centre for Child Health, University of Basel | BRCCH_MIP | Shinichi Sunagawa Emma Slack |
| Agence Nationale de la Recherche | ANR-21-CE45-0015 | Claude Loverdo |
| Centre National de la Recherche Scientifique | MITI CNRS AAP adaptation du vivant à son environnement | Claude Loverdo |
| Schweizerischer Nationalfonds zur Förderung der Wissenschaftlichen Forschung | 40B2-0_180953 | Emma Slack |

| Funder | Grant reference number | Author |
|---|---|---|
| Schweizerischer Nationalfonds zur Förderung der Wissenschaftlichen Forschung | 310030_185128 | Emma Slack |
| Agence Nationale de la Recherche | ANR-20-CE30-0001 | Claude Loverdo |

The funders had no role in study design, data collection and interpretation, or the decision to submit the work for publication.

## Author contributions

Daniel Hoces, Conceptualization, Formal analysis, Validation, Investigation, Visualization, Methodology, Writing – original draft, Writing – review and editing; Giorgia Greter, Investigation, Methodology, Writing – review and editing; Markus Arnoldini, Conceptualization, Formal analysis, Visualization, Methodology, Writing – review and editing; Melanie L Stäubli, Anna Sintsova, Formal analysis, Methodology, Writing – review and editing; Claudia Moresi, Sara Berent, Investigation, Writing – review and editing; Isabel Kolinko, Resources, Methodology, Writing – review and editing; Florence Bansept, Aurore Woller, Methodology, Writing – review and editing; Janine Häfliger, Eric Martens, Resources, Writing – review and editing; Wolf-Dietrich Hardt, Resources, Funding acquisition, Writing – review and editing; Shinichi Sunagawa, Supervision, Funding acquisition, Methodology, Writing – review and editing; Claude Loverdo, Conceptualization, Resources, Formal analysis, Supervision, Funding acquisition, Validation, Investigation, Visualization, Methodology, Writing – review and editing; Emma Slack, Conceptualization, Resources, Formal analysis, Supervision, Funding acquisition, Investigation, Methodology, Writing – original draft, Writing – review and editing

## Author ORCIDs

Daniel Hoces ![ORCID] http://orcid.org/0000-0002-1451-5166
Florence Bansept ![ORCID] http://orcid.org/0000-0003-0562-9222
Wolf-Dietrich Hardt ![ORCID] http://orcid.org/0000-0002-9892-6420
Shinichi Sunagawa ![ORCID] http://orcid.org/0000-0003-3065-0314
Claude Loverdo ![ORCID] http://orcid.org/0000-0002-0888-1717
Emma Slack ![ORCID] http://orcid.org/0000-0002-2473-1145

## Ethics

All animal experiments were performed with approval from the Zürich Cantonal Authority under license number ZH120/19.

## Decision letter and Author response

Decision letter https://doi.org/10.7554/eLife.81212.sa1
Author response https://doi.org/10.7554/eLife.81212.sa2

# Additional files

## Supplementary files

• Supplementary file 1. Genomic variants (SNPs, small insertions, and deletions) in *B. theta* strains whole-genome sequences. Whole-genome sequencings of all barcoded and untagged strains (WT and acapsular) were mapped against *Bacteroides thetaiotaomicron* strain VPI 5482 genome (CP092641.1) to identify genetics variants. Data are included in *Figure 4—source data 1*.

• MDAR checklist

## Data availability

Relevant numerical source data for figures and figure supplements has been provided. Raw sequencing data accessed on ENA (https://www.ebi.ac.uk/ena/browser/home) under Project ID PRJEB57876 and PRJEB53981. Raw data and code used for generating all figures in this publication are made available in a curated data archive at ETH Zurich (https://www.research-collection.ethz.ch/) under the https://doi.org/10.3929/ethz-b-000557179.

The following datasets were generated:

| Author(s) | Year | Dataset title | Dataset URL | Database and Identifier |
|---|---|---|---|---|
| Slack E | 2022 | Population dynamics of Bacteroides thetaiotaomicron during the early colonization of the murine gut | https://www.ebi.ac.uk/ena/browser/view/PRJEB57876 | European Nucleotide Archive, PRJEB57876 |
| Burga H, Alexander D | 2022 | Fitness advantage of Bacteroides thetaiotaomicron capsular polysaccharide is dependent on the resident microbiota | https://doi.org/10.3929/ethz-b-000557179 | ETH Zurich Research Collection, 10.3929/ethz-b-000557179 |

The following previously published dataset was used:

| Author(s) | Year | Dataset title | Dataset URL | Database and Identifier |
|---|---|---|---|---|
| ETH ZURICH | 2022 | Metabolic reconstitution by a gnotobiotic microbiota varies over the circadian cycle | https://www.ebi.ac.uk/ena/browser/view/PRJEB53981 | European Nucleotide Archive, PRJEB53981 |

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

## Appendix 1

### Supplementary Methods

### 1. Colonization

#### 1.1 Initial bacteria number

Let us denote $C_0$ the bacterial concentration in the prepared solution. Then if there is a volume $V$ of solution, there are $N = C_0 V$ bacteria. Then, the probability to have taken $n_0$ bacteria in a volume $v_0$ is

$$p\left(n_0\right) = BinomialDistribution\left(N, v_0/V\right) = \left(\frac{v_0}{V}\right)^{n_0} \left(1 - \frac{v_0}{V}\right)^{N-n_0} \frac{N!}{\left(N - n_0\right)! n_0!} \tag{A1}$$

In the limit of $N = cV$ large and $v_0 \ll V$,

$$p\left(n_0\right) \simeq PoissonDistribution\left(Nv_0/V\right) = \frac{\left(\frac{Nv_0}{V}\right)^{n_0} \exp\left(-\frac{Nv_0}{V}\right)}{n_0!} \tag{A2}$$

#### 1.2 Colonization probability via the loss: Base theory

Let us define $\beta$ as the probability for each bacterium to get to the cecum alive, and then have its lineage survive until measurement. There is a priori no interaction early on between incoming bacteria as their concentration is initially low enough to limit the competition between them. Then, if started with an average of $n_0$ bacteria (Poisson distributed), the probability for a given barcode to be absent at measurement time is the zero of the Poisson distribution of average $\beta n_0$, and thus

$$p_{loss} = \exp\left(-\beta n_0\right) \tag{A3}$$

Then as $n_0$ is estimated via the concentration and volume of the inoculum, and $p_{loss}$ is best estimated via the number of barcodes lost divided by the total number of barcodes, $\beta$ is estimated as

$$\beta \simeq \frac{-\log\left(n_{losttags}/n_{tags}\right)}{n_0} \tag{A4}$$

#### 1.3 Colonization probability via the loss: Handling different starting inoculum sizes and calculating best estimate

The barcoded bacteria are not necessarily in equal concentrations in the inoculum and the data from several experiments with different inoculums need to be combined. Let us define $w$ the number of barcoded multiplied by the number of mice. For each of these $w$, there were $n_i$ barcoded bacteria in the inoculum, and we define $l_i$ as 1 of the barcodes was lost, 0 otherwise. Then the estimate of $\beta$ is $\beta$, which maximizes $proba\left(l_1, ...., l_w\right)$ the likelihood to observe $\{l_1, l_2, ...l_w\}$; and it is the same as maximizing the log likelihood. As for each barcoded bacterium the process will be considered as independent, it will then be simply the maximization of

$$LL = \sum_{i=1}^{w} \log\left(\left(\exp\left(-\beta n_i\right)\right)^{l_i} \left(1 - \exp\left(-\beta n_i\right)\right)^{1-l_i}\right) \tag{A5}$$

This is the same as maximizing the following expression:

$$LL = \sum_{i=1}^{w} \left(-l_i \beta n_i + \left(1 - l_i\right) \log\left(1 - \exp\left(-\beta n_i\right)\right)\right) \tag{A6}$$

Let us note $x = -\beta$.

$$\frac{dLL}{dx} = \sum_{i=1}^{w} n_i \left(l_i - \left(1 - l_i\right) \frac{\exp\left(xn_i\right)}{1 - \exp\left(xn_i\right)}\right) \tag{A7}$$

The value of $x$ that makes this expression zero is found numerically and enables to infer $\beta$, while combining data with different initial number of bacteria in the inoculum coming from different mice.

## 1.4 Colonization probability via the loss: Handling different starting inoculum sizes and calculating confidence interval

To calculate a confidence interval, it is useful to obtain an estimate of the probability that a given $\beta$ is the true value knowing the given observations, which can be denoted $p\left(\beta|observations\right)$. Therefore, we used a Bayesian approach, $p\left(\beta|observations\right) = p\left(observations|\beta\right)p\left(\beta\right)/p\left(observations\right)$. In this expression, $p\left(observations|\beta\right)$ is the probability to observe $\{l_1, l_2, ...l_w\}$ for a given $\beta$, so it is actually $\exp\left(LL\right)$. Then the prior on $p\left(\beta\right)$ has to be chosen. As it may be frequent to have a very low probability of survival, a flat prior for $p\left(\log\left(\beta\right)\right)$ may be better than a flat prior for $p\left(\beta\right)$. Then a similar reasoning can be done, $p\left(\log\left(\beta\right)|observations\right) = p\left(observations|\log\left(\beta\right)\right)p\left(\log\left(\beta\right)\right)/p\left(observations\right)$, and with a flat prior for $p\left(\log\left(\beta\right)\right)$, $p\left(\log\left(\beta|observations\right)\right) \propto p\left(observations|\log\left(\beta\right)\right)$. Then $\exp(LL(\log(\beta)))$ renormalized by its integral for $\log\left(\beta\right)$ gives an estimate of the distribution of probability of inference of $\log\left(\beta\right)$. The exponential of the average of $\log\left(\beta\right)$ on this distribution is very close to the value of $\beta$ maximizing $LL'$. This distribution is very close to a Gaussian distribution (see **Appendix 1—figure 1**). Then the mean $\log\left(\beta\right)$ ± twice the standard deviation of $\log\left(\beta\right)$ on this distribution gives a 95% confidence interval.

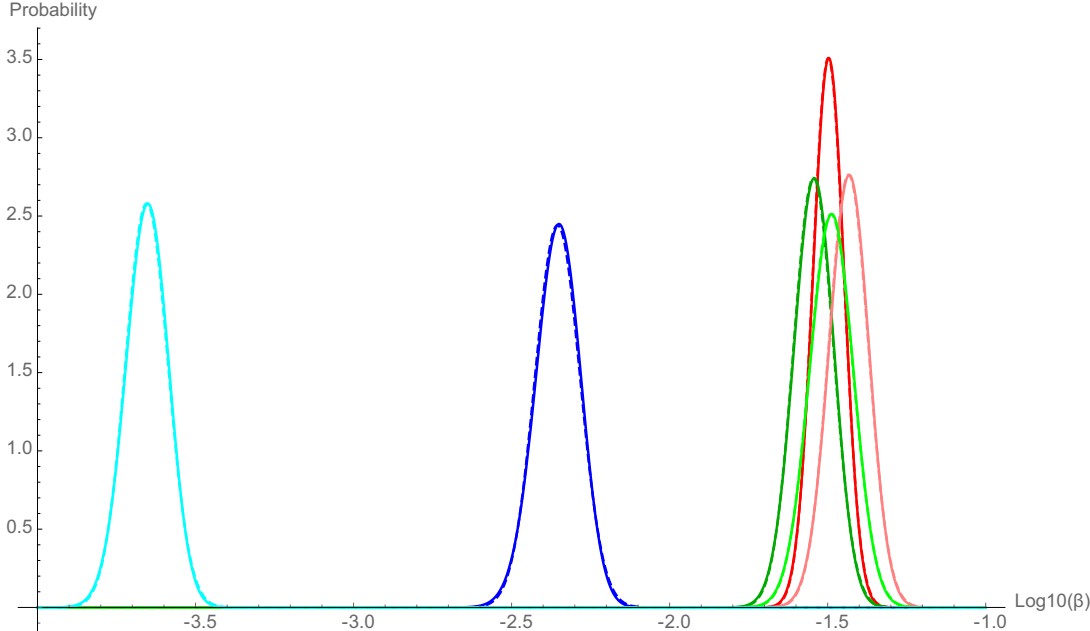

**Appendix 1—figure 1.** Distribution of the probability estimation of $\boldsymbol{log}_{10}\left(\boldsymbol{\beta}\right)$. The colors represent experiments: low-complexity microbiota (LCM) WT (red), Oligo WT (green), specific pathogen free (SPF) WT (blue), LCM acapsular (pink), Oligo acapsular (light green), and SPF acapsular (light blue). Solid lines represent the numerical result for the probability distribution of $\beta$ using renormalized **equation (6)**, the dashed lines represent the normal distribution with the same mean $log_{10}\left(\beta\right)$ and same variance than the numerical distribution.

Note that if we had chosen a flat prior on $p\left(\beta\right)$ rather than $p\left(\log\left(\beta\right)\right)$, the result would have been very similar, though with a mean slightly further away from $\beta$ maximizing $LL'$, and the distribution would be less similar to a normal distribution.

## 1.5 Colonization probability via the loss: Results

The results here are the values of $\beta$ maximizing the log likelihood, with the confidence interval as explained in previous section:

*Continued on next page*

| | Low-complexity microbiota | Oligo | Specific pathogen free |
|---|---|---|---|
| WT | $10^{-1.50}[-1.60:-1.40]$ | $10^{-1.54}[-1.67:-1.42]$ | $10^{-2.35}[-2.50:-2.21]$ |
| Acapsular | $10^{-1.43}[-1.56:-1.31]$ | $10^{-1.49}[-1.63:-1.35]$ | $10^{-3.65}[-3.79:-3.52]$ |

In principle, as we make multiple comparisons, the confidence interval should be adjusted, but here, even without correction, the LCM/oligo mice for both WT and acapsular are not significantly different (adjusting for multiple comparisons would make them even less distinguishable). In the case of the SPF mice, the resulting $\log_{10}(\beta)$ is more than 10 standard deviations away for the WT bacteria, and more than 30 standard deviations for the acapsular bacteria, relative to LCM/Oligo mice. Within the SFP mice, the difference between WT and acapsular bacteria is more than 18 standard deviations. Because the differences are very large, even when adjusting for multiple comparisons, they would remain significant.

## 1.6 Colonization probability via the loss: Interpretation of the apparent loss probability (initial loss and subsequent loss)

### 1.6.1 Principle
What is obtained here is the probability for a bacterium in the inoculum to have seeded a lineage still present at measurement. For the lineage to be still present, it needs to

- make it alive to the cecum. Let us denote this probability $q$.
- escape stochastic fluctuations, mainly during the initial rounds of reproduction.

If there is little subsequent loss, then $\beta \simeq q$.

### 1.6.2 Minimal model
Here, we assume that each bacterium has a probability $q$ of establishing, and then its lineage grows at a rate $r$ and is lost at a rate $c$. After some time, carrying capacity is reached. Here, we assume the population is large, and thus stochastic loss will be negligible at this point. Therefore, as the stochastic effects occur when the population size is small, then it is legitimate to focus on this step and neglect saturation.

If a bacterium survives the first step (probability $q$), and its lineage reaches size $n$ at time time $t$ with probability $p_1(n,t)$, given bacteria replicate at a rate $r$ and are cleared at a rate $c$,

$$\frac{dp_1(n,t)}{dt} = -n(c+r)p_1(n,t) + c(n+1)p_1(n+1,t) + r(n-1)p_1(n-1,t). \tag{A8}$$

Multiplying this equation by $z^n$, and summing all equations from $n=0$ (with $p_1(-1,t)=0$) to infinity, and defining the generating function:

$$g_1(z,t) = \sum_{n=0}^{\infty} p_1(n,t)z^n, \tag{A9}$$

$$\frac{\partial g_1(z,t)}{\partial t} = (1-z)(c-rz)\frac{\partial g_1(z,t)}{\partial z}. \tag{A10}$$

Using the method of the characteristics, and with the condition that $g_1(z,t=0)=z$, as by definition $p_1(n=1,t=0)=1$ and $p_1(n\neq1,t=0)=0$, it can be calculated that

$$g_1(z,t) = \frac{(rz-c)\exp(-(r-c)t)+c(1-z)}{(rz-c)\exp(-(r-c)t)+r(1-z)} \tag{A11}$$

Then, defining $p(n,t)$ the probability to observe a lineage of size $n$ at time $t$; starting with 1 bacterium, before the bottleneck, and $g(z,t)$ the associated generating function, then $p(0,t)=1-q+p_1(0,t)$, and $p(n>0,t)=qp_1(n,t)$, which leads to $g(z,t)=1-q+qg_1(z,t)$, and finally

$$g(z,t) = 1 - q\frac{(r-c)(1-z)}{(rz-c)\exp(-(r-c)t)+r(1-z)} \tag{A12}$$

The resulting loss probability is

$$p_{loss} = g\left(0, t\right) = 1 - q\frac{r - c}{r - c\exp\left(-\left(r - c\right)t\right)} \xrightarrow{\left(r - c\right)t \gg 1} 1 - q\left(1 - \frac{c}{r}\right) \tag{A13}$$

### 1.6.3 Model with delay

As shown in **Figure 3—figure supplement 2**, we observed a delay in the start of growth in the mouse gut, during which time bacteria will continue to be lost due to flow but will not replicate. Therefore, in a model with a fixed delay, it would be the same, except with $q\exp\left(-c\tau\right)$ instead of $q$.

### 1.7 Colonization probability via the loss: Optimal $n_0$

Here $\beta$ is the probability for the lineage of one bacterium, present in the inoculum, to be found at measurement time. It is a combination of making it alive to the cecum, and not being lost afterwards. It is the quantity to estimate.

The mean number of bacteria $n_0$ of a given barcode in the inoculum can be controlled, and thus the question is what is the optimal $n_0$ to use, which will give the most accurate $\beta$ estimate. For instance, one can perform preliminary experiments to assess the order of magnitude of the colonization probability and then tune $n_0$ for better accuracy.

$w$ is the total number of different barcodes measured (the number of different bacterial barcodes in each mouse multiplied by the number of mice). We note $n_l$ the number of barcodes among them that are lost in the experiment, that is, that are not detected at the end of the experiment.

As seen in Section 1.2 at **equation (4)**, the simplest $\beta$ estimate is $-\log\left(n_l/w\right)/n_0$ .

If in the inoculum there are an average of $n_0$ bacteria of each barcode, Poisson distributed (see Section 1.1), the probability that $n_l$ lineages are lost is

$$p\left(n_l\right) = \exp\left(-\beta n_0\right)^{n_l}\left(1 - \exp\left(-\beta n_0\right)\right)^{w - n_l}\frac{w!}{\left(w - n_l\right)!n_l!} \tag{A14}$$

### Probability of all lost and none lost

One issue is that there may be experiments with $n_l = 0$ (in which case the previous expression is not well defined) or $n_l = w$ (in which case the estimate is that $\beta = 0$, i.e., no bacteria survive).

The probability to observe the loss of all barcodes is

$$p\left(all\ lost\right) = \exp\left(-\beta n_0 w\right) \tag{A15}$$

The probability to observe no barcode loss is

$$p\left(no\ loss\right) = \left(1 - \exp\left(-\beta n_0\right)\right)^w \tag{A16}$$

The sum of these two probabilities is minimized for $\exp\left(-\beta n_0\right) = 1/2$, i.e., $\beta n_0 = \log\left(2\right) \simeq 0.69$.

In the case for which we do not estimate $\beta$ when either all or no barcodes are lost, the expected error in the estimate can be written as

$$\frac{\left\langle\left(\beta_{estimated} - \beta_{true}\right)^2\right\rangle}{\beta_{true}^2} = \frac{\sum_{n_l=1}^{w-1}\left(\frac{-\log\left(n_l/w\right)}{\beta n_0} - 1\right)^2\exp\left(-\beta n_0\right)^{n_l}\left(1 - \exp\left(-\beta n_0\right)\right)^{w - n_l}\frac{w!}{\left(w - n_l\right)!n_l!}}{1 - \left(1 - \exp\left(-\beta n_0\right)\right)^w - \exp\left(-\beta n_0 w\right)} \tag{A17}$$

### Numerical results

As shown in **Appendix 1—figure 2**, the maximum probability of obtaining an estimate is for $\beta n_0 = \log\left(2\right) \simeq 0.69$, as expected. For this value, the expected error in the estimate is relatively small. The error slowly increases for decreasing $\beta n_0$ , is minimal for $\beta n_0$ somewhat larger than 0.69, but then with increasing $\beta n_0$ , the probability to obtain no estimate quickly increases.

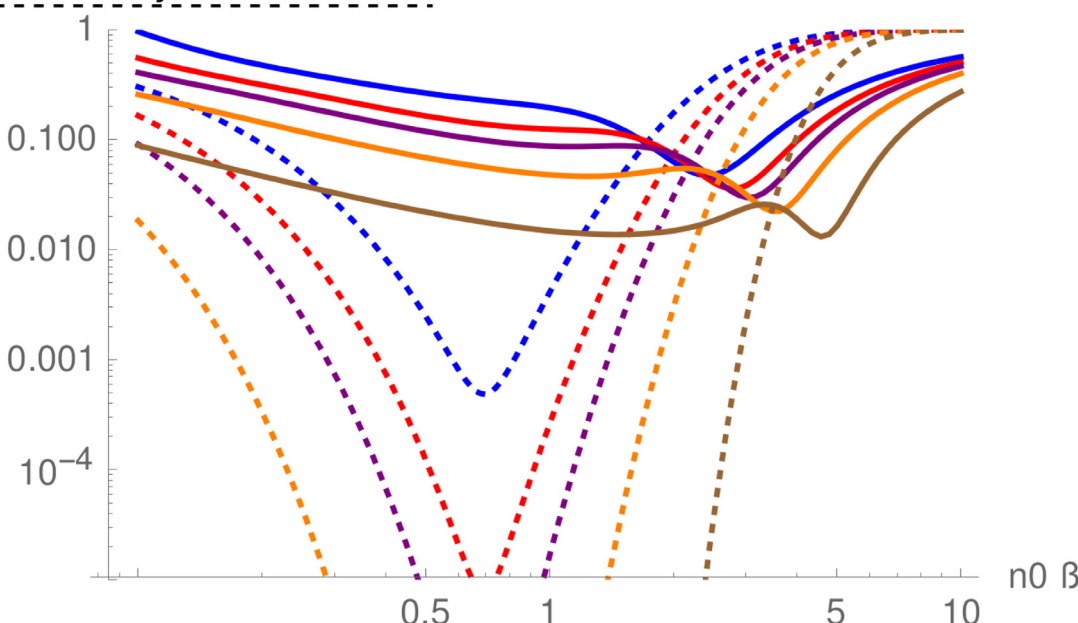

**Appendix 1—figure 2.** Mean relative square error in the estimate of $\beta$, as a function of $n_0\beta$. The solid-colored lines are the expression (17) for $w = 12$ (blue), 18 (red), 24 (purple), 40 (orange), and 120 (brown). The dotted lines are the probability that no estimate is given, either because all the tags are lost, or no tag is lost. They are the sum of **equations (15) and (16)**.

## Conclusion

The value of $n_0$ leading to the best $\beta$ estimates (likely possibility of obtaining an estimate and minimized relative difference between the estimate and the true value) is such that $\beta n_0$ is of the order of 1.

## 1.8 Colonization probability using the variance in the barcoded population sizes: Based theory

### 1.8.1 Simple bottleneck

Here, we assume that each bacterium has a probability $\beta$ of establishing and then grows deterministically. If the growth rate is *net* and the total population size remains below the carrying capacity at $t$, bacteria grow by a factor of $\exp\left(net \times t\right)$ after the bottleneck.

Then

$$p_{loss} = 1 - \beta \tag{A18}$$

$$var = \left(\beta\left(1-\beta\right)^2 + \left(1-\beta\right)^2\beta\right)\exp\left(2nett\right) = \beta\left(1-\beta\right)\exp\left(2nett\right) \tag{A19}$$

The relative variance, which is not sensitive to the fact that different mice have different final carrying capacity, is

$$var_{rel} = \frac{var}{\beta^2\exp\left(2nett\right)} = \frac{1-\beta}{\beta} \xrightarrow{\beta \ll 1} \frac{1}{\beta} \tag{A20}$$

Here $var_{rel} = 1/\left(1 - p_{loss}\right)$.

### 1.8.2 Bottleneck, replication rate, and loss rate

Similar to the previously described model, we assume that each bacterium has a probability $q$ of establishing, and then grows at a rate $r$ and is lost at a rate $c$, stochastically. As seen in the previous

section, the corresponding generating function for the population size at time $t$, starting from one bacterium before the bottleneck, is

$$g\left(z,t\right) = 1 - q\frac{\left(r - c\right)\left(1 - z\right)}{\left(rz - c\right)\exp\left(-\left(r - c\right)t\right) + r\left(1 - z\right)} \tag{A21}$$

and the resulting loss probability is

$$p_{loss} = g\left(0,t\right) \overset{(r-c)t \gg 1}{\to} 1 - q\left(1 - \tfrac{c}{r}\right) \tag{A22}$$

The resulting variance on the population size is

$$var = \left(\frac{\partial^2 g}{\partial z^2} + \frac{\partial g}{\partial z} - \left(\frac{\partial g}{\partial z}\right)^2\right)_{z=1} \tag{A23}$$

$$var = q^2 e^{2(r-c)t}\left(\frac{1}{q\left(1 - \tfrac{c}{r}\right)}\left(2 - \left(1 + \tfrac{c}{r}\right)e^{-(r-c)t}\right) - \frac{1 - \tfrac{c}{r}}{1 + \tfrac{c}{r}}\right) \tag{A24}$$

$$var \overset{(r-c)t \gg 1}{\to} q^2 e^{2(r-c)t}\left(\frac{2}{q\left(1 - \tfrac{c}{r}\right)} - \frac{1 - \tfrac{c}{r}}{1 + \tfrac{c}{r}}\right) \tag{A25}$$

$$var_{rel} = \frac{var}{\left(q\exp\left(\left(r - c\right)t\right)\right)^2} = \frac{2}{q\left(1 - c/r\right)} - \frac{1 - c/r}{1 + c/r} \overset{q \ll 1}{\to} \frac{2}{q\left(1 - c/r\right)} \tag{A26}$$

Here $var_{rel} = 2/\left(1 - p_{loss}\right)$.

### 1.8.3 Different initial number of bacteria

In the limit where the initial number of bacteria of the different barcodes are of the same order of magnitude, and with $h$ the number of different barcodes in one mouse, the variance on the proportions is

$$var\left(p\right) = \tfrac{1}{h-1}\sum\left(p_i - 1/h\right)^2 = \tfrac{1}{h-1}\sum\left(\frac{n_i}{\sum n_j} - \tfrac{1}{h}\right)^2 \tag{A27}$$

After some calculations, we find

$$\left\langle var\left(p\right)\right\rangle - varp_0 \simeq \frac{1}{h\sum n_{j,0}}\frac{var_1}{m_1^2} \tag{A28}$$

with $varp_0$ the variance in proportions in the inoculum, $\sum n_{j,0}$ the total number of different barcodes in the inoculum, and $var_1/m_1^2$ the relative variance starting from one bacterium, which is approximately $\frac{2}{q(1-c/r)}$ and thus expected to be approximately equal to $2/\beta$, with $\beta$ the apparent survival probability.

Then

$$\frac{var_1}{m_1^2} \simeq h\sum n_{j,0}\left(\left\langle var\left(p\right)\right\rangle - varp_0\right) \tag{A29}$$

### 1.8.4 Procedure

The procedure is then to estimate $\frac{var_1}{m_1^2}$ for each mouse using this equation, and then average the results to obtain $\left\langle \frac{var_1}{m_1^2}\right\rangle$, and

$$\beta \simeq \frac{2}{\left\langle var_1/m_1^2\right\rangle}. \tag{A30}$$

There is one mouse (for the acapsular in SFP) for which all the barcodes were lost. We remove this mouse from the analysis but keep the 14 others so that the bias is likely minimal.

Then twice the standard error (the standard error is the standard deviation divided by the square root of the number of mice for each condition, reflecting that the more mice, the better the average

is determined) is used for the confidence interval around $\left\langle \frac{var_1}{m_1^2} \right\rangle$, which bounds are then used to define the bounds of the confidence interval for the estimate of $\beta$ via the variance.

## 1.9 Colonization probability using the variance in the sizes of barcoded populations: Results and comparison with barcode loss

|  | Low-complexity microbiota | **Oligo** | Specific pathogen free |
|---|---|---|---|
| WT | $10^{-1.66}[-1.78: -1.51]$ | $10^{-1.85}[-2.05: -1.51]$ | $10^{-2.42}[-2.62: -2.02]$ |
| Acapsular | $10^{-1.90}[-2.07: -1.63]$ | $10^{-1.61}[-1.72: -1.47]$ | $10^{-3.85}[-3.99: -3.66]$ |

*Appendix 1—figure 3* compares the loss and the variance method. Overall:
- Both methods give the same orders of magnitude.
- Both methods show that in this dataset, there is no significant difference for all the experiments in LCM and oligo mice; while WT in SPF mice has a tighter bottleneck (though not significant in the variance method), and acapsular in SPF mice even more.
- Overall, the variance method results in smaller $\beta$ estimates. The variance method is more sensitive to the model used, and approximations made in the minimal model may explain this systematic difference.

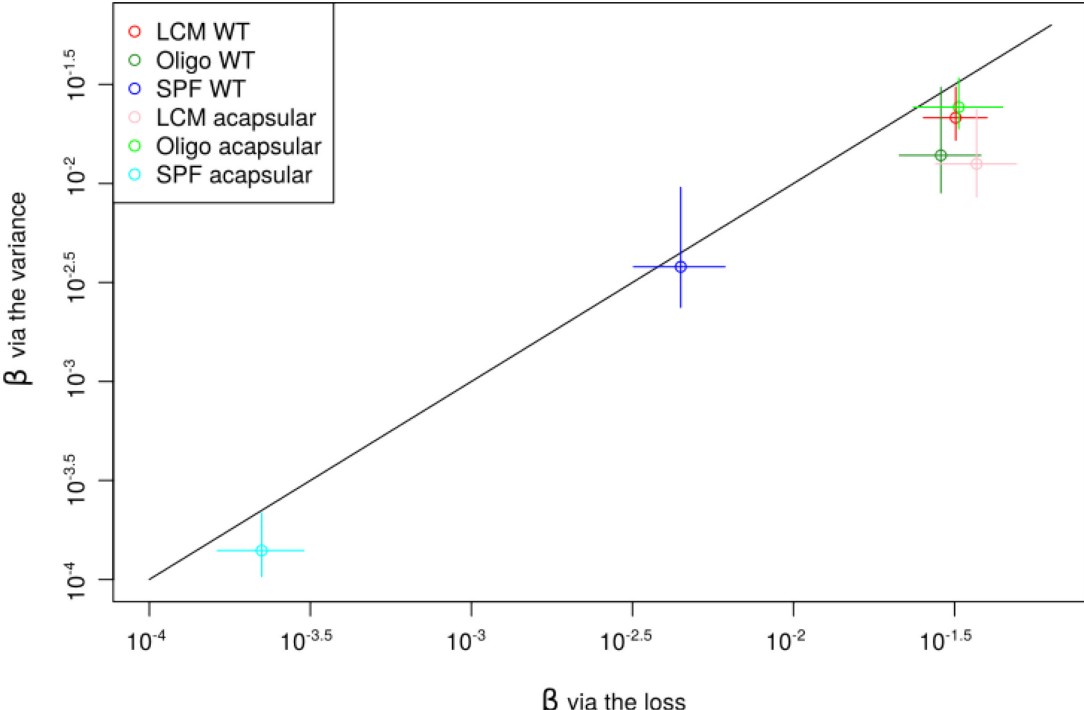

**Appendix 1—figure 3.** Comparison between the lineage survival probability estimated via the loss method and via the variance method. The colors represent experiments: low-complexity microbiota (LCM) WT (red), Oligo WT (green), specific pathogen free (SPF) WT (blue), LCM acapsular (pink), Oligo acapsular (light green), and SPF acapsular (light blue). Circles represent best estimation and bars the error. The error bars are given via the direct study of the estimation probability for the loss method via the standard error between mice for the variance method.

Note that if there is a fixed delay $\tau$ before growing, then the equations are the same, except for $q$ which is replaced by $q\exp(-c\tau)$. Then the relation between the $\beta$ estimate via the loss or the variance remains the same, and $\beta$ is an apparent survival probability, considering delay additionally to the initial bottleneck.

## 1.10 Experimental noise

There are incertitudes on measurements:

- To estimate overall bacterial concentration in the solution from which the inoculum is taken from, a number of colonies $n$ is counted, and so we expect a typical relative error in the concentration of $1/\sqrt{n}$ .
- The relative proportion of the different barcoded strains are analyzed by qPCR. In an experiment, three replicates were measured for each of the barcoded strains, and the standard deviation in the counts was 0.22. Note that $2^{0.22} \simeq 10^{0.066}$ , thus if the incertitude in the counts is ±0.22, then if the number/concentration estimated using the qPCR is expressed as a power of 10, the incertitude will be ±0.066 in the exponent.

For the loss method, the end point measurement is the absence/presence of a barcoded strain, which is not sensitive to the qPCR incertitudes (at least at the small level of qPCR noise of the experiments). However, the calculations actually estimate $\beta n_i$ , thus errors in the estimate of $n_i$ , the initial expected number of bacteria for each barcoded strain, will affect the estimate of $\beta$.

For instance, the typical number of colonies counted for checking the initial concentration from which $n_0$ is calculated is of the order of 40, resulting in a typical relative error of $1/\sqrt{40}$ , that is, about 15%. Then overestimating $n_0$ by 15% will result in underestimating $\beta$ by about 15%. The relative error is not biased, it will just increase the incertitude. ±15% will corresponds to approximately ±0.06 in the $log10\left(\beta\right)$ . This is if all data was from the same inoculum. Actually, for each condition, the data is pooled from different experiments, with different starting inoculums, with uncorrelated incertitudes on the initial number of barcoded bacteria (2–5 different starting inoculums depending on the condition). Then this incertitude is smaller than the incertitude as calculated previously. As a consequence, while it would somewhat increase the confidence interval, taking into account this source of incertitude has a small impact, and it is not included in our main results for simplicity.

The impact of the qPCR incertitude on the loss method is smaller. Indeed, the incertitude due to qPCR counts for one barcoded strain is of the same order of magnitude as the incertitude on the total $n_0$ discussed in previous paragraph; but then for each inoculum there are six barcoded strains, and the total number of barcoded bacteria is fixed, thus the overestimates and underestimates will almost compensate, and the resulting incertitude will be small compared to the incertitude on the total $n_0$ .

For the variance method, calculations show that the expressions can be modified to consider the incertitude on $n_0$ , and with $\sigma$ the standard deviation in noise for the number of counts,

$$\langle var_p \rangle \simeq var_{p0} + \frac{var_1}{m_1^2} \frac{1}{h^2 \langle n_0 \rangle} + \frac{1}{h^2 \langle n_0 \rangle} + \frac{2\log\left(2\right)^2 \sigma^2}{h^2}. \tag{A31}$$

With the experimental values, this correction is very small and thus is not included in the main results for simplicity.

## 1.11 No tag bias in the colonization results

Bacteria with different tags may grow at different rates and this may bias the results on colonization probabilities. We checked that in vitro, the growth rates between the different barcoded bacteria are not significantly different. In this section, we show that if we were analyzing only a subset of the barcoded bacteria, the results on the colonization probabilities would be very similar, showing the robustness of the results.

In fact, in the case of the loss method, as the readout (absent or present) is sensitive only to the early dynamics, and not later growth (as soon as the number of bacteria becomes large enough), we do not expect any effect of tag fitness.

Indeed, as shown in **Figure 2—figure supplement 5B**, while the results using only one tag are noisier, they are in line with the estimate using all the data, and there is no sign of a tag having a consistent bias.

Several tags for each mouse are necessary for the variance method. Removing a tag with a substantially different growth rate than the others would decrease the variance, and thus increase the $\beta$ estimate. In **Figure 2—figure supplement 5C**, removing one barcode has very little effect on the estimates.

## 2. Growth curves

### 2.1 Net growth rate

If the bacteria grow at rate $r$ and are cleared at rate $c$, the slope of the growth curve, with the log of the bacterial concentration represented as function of time, will be the net growth rate $net = r - c$. We remove the first points (which may also contain bacteria from the inoculum that are passively carried and had not really settled in the cecum); and the last points, at which the bacterial concentration is close to carrying capacity, and thus the population is not growing any more.

### 2.2 Linear growth fits

See *Figure 3—figure supplement 2* for the fits for either WT or acapsular, growing in mice with OligoMM12 microbiota, and *Figure 3C* for the values. We find a significant difference (p<0.01), at 0.50/hr (WT) vs. 0.40/hr (acapsular).

In the data of *Figure 2—figure supplement 2* for the WT in Oligo and LCM mice, there were no significant differences between the growth rates in these microbiotas, which was expected. A linear fit on WT bacteria in three SPF mice gives a net growth rate of 0.26, 0.44, and 0.40/hr, that is, and average of 0.37 ± 0.09/hr. The net growth rate lower by 0.13/hr is, with the error bars, actually consistent with a higher turnover in SPF mice, 0.23/hr instead of 0.13/hr.

### 2.3 Delay to start exponential growth: Base theory

As observed in the growth curves in *Figure 3—figure supplement 2*, there seem to be a delay in growth. To quantify this, we analyzed the growth curves as shown in *Figure 3—figure supplement 2A*.

With a survival probability $q$, a delay $\tau$ with clearance rate $c$, and then regrowth at rate $net$, the equation for the part of the curve with growth is $n_0 q \exp\left(-c\tau\right) \exp\left(net\left(t - \tau\right)\right)$. For the colonization, $q_{app} = q \exp\left(-c\tau\right)\left(1 - c/r\right)$. Then the equation for the curve is $n_0 q_{app} \exp\left(net\left(t - \tau\right)\right) / \left(1 - c/r\right)$. When fitting the curve of the growth part, the intercept $I$ (=concentration value when $t = 0$) is then $n_0 q_{app} \exp\left(net\left(-\tau\right)\right) / \left(1 - c/r\right)$. Then $\tau$ can be estimated as

$$\tau = \log\left(\left(n_0 q_{app}\right) / \left(I\left(1 - c/r\right)\right)\right) / net \tag{A32}$$

$I$ and $net$ comes from the fit, $n_0$ is the inoculum size, $q_{app}$ can be taken from the colonization experiments.

### 2.4 Delay to start exponential growth: Bacterial concentration in feces vs. absolute numbers in cecum

Note that the reasoning of the previous subsection is for the absolute number of bacteria in the cecum, whereas the growth curves are obtained from bacterial concentration in feces. This is not an issue if we only measure $net$, but to estimate the delay, we need to convert $n_0$ of the inoculum in effective $c_0$ for the feces. To do so, we need both the cecum mass, and the feces to cecum concentration factor.

If the bacterial concentration in the feces $c$ is $f$ times higher than in the cecum (mostly due to water absorption), and that the cecum has mass $m$, then the absolute number in the cecum $n$ is such that $c = f \times n/m$. Then if we look at the time course in the feces concentration, to convert the inoculum absolute $n_0$ in equivalent concentration in feces, $c0 = f \times n_0/m$. Thus, we need to determine $m$ and $f$.

In experiments where the total cecum mass in oligo mice was measured (*Hoces et al., 2022*), the mean mass was $1.5 \pm 0.2g$. Note that an error of 50% in the mass for a net growth rate of 0.5/hr would be around $log\left(1.5\right)/0.5 \simeq 0.8h$.

Now let us estimate $f$. We expect a higher concentration in feces relative to the cecum due to water absorption. However, in experimental data, which is noisy, we find that the average ratio of concentration of feces relative to the cecum very close to 1.

## 2.5 Fitting the delay

### Process

$c_0$ is the effective $n_0$ multiplied by $f/m \simeq 0.7$. $q_{app}$ is taken as $\simeq 10^{-1.5}$ from the colonization experiment. We estimate with the expression (32), taking $c = 0.13$ for the WT, and $c = 0.23$ for the acapsular (considering that $net$ is smaller for the acapsular, likely because of a higher clearance rate).

### Results

See *Figure 3—figure supplement 2* for looking at all the fits, and *Figure 3D* for the resulting delays. We find a significant difference between WT and acapsular, with an average delay of 3.2 hr for the WT and 7.7 hr for the acapsular (p=0.02).

Note that the delay for the WT could come just from the fact that it takes time for bacteria to physically go from the oesophagus to the ceum. The delay for the acapsular could come from a combination of such a transport delay, +a delay to resume growth.

These 4.5 hr of delay in acapsular relative to the WT is at the limit to keep coherence with the colonization probabilities. With such a delay, and for the higher loss rate of acapsular, that we estimate at 0.23/h, then the expected difference in apparent colonization probability (assuming the same initial survival probability) is $\simeq 10^{0.5}$, which is almost the double than what is coherent with the values found and their incertitudes. Note that several factors could influence this result: if we underestimate $f$, we overestimate the relative delay between WT and acapsular. If we overestimate the clearance difference between WT and acapsular, we also overestimate the relative delay between WT and acapsular.

## 2.6 Growth dynamics with competition

We propose here to fit the growth data of an experiment in which WT and acapsular were given in a 1:1 ratio.

As the WT is quite more concentrated in the feces relative to the acapsular when approaching carrying capacity, it is fair to assume its growth is independent of the acapsular one, and use logistic growth,

$$\frac{dn}{dt} = r \left(1 - \frac{n}{K_{max}}\right) n\left(t\right) - cn\left(t\right) \tag{A33}$$

with $K_{max}$ the maximum carrying capacity, $r$ the maximal growth rate, and $c$ the loss rate. Then, after some calculations, and with $K = K_{max}\left(1 - c/r\right)$ the effective carrying capacity, and $r_{net} = r - c$ the effective growth rate,

$$n\left(t\right) = \frac{K}{1 + \left(\frac{K}{n_0} - 1\right) \exp\left(-r_{net}t\right)}. \tag{A34}$$

For the acapsular, the dynamics for the acapsular, as acapsular and WT are likely to be limited by the same nutrients:

$$\frac{dA}{dt} = r_a \left(1 - \frac{A+W}{K_{max}}\right) A - c_a A. \tag{A35}$$

In this experiment, either $\left(A + W\right)/K_{max} \ll 1$, or $A \ll W$, thus

$$\frac{dA}{dt} \simeq r_a \left(1 - \frac{W}{K_{max}}\right) A - c_a A, \tag{A36}$$

which with equation (*A34*), leads to

$$A\left(t\right) = a_0 \exp\left(net_a t\right) \left(\frac{\frac{K}{w_0}}{\exp\left(net_w t\right) - 1 + \frac{K}{w_0}}\right)^{rr} \tag{A37}$$

with $K = K_{max}\left(1 - c_w/r_w\right)$, $a_0$ the initial concentration in acapsular, $w_0$ the initial concentration in wild type, $net_a = r_a - c_a$ the net maximal growth rate of the acapsular, and $rr = r_a/r_w$ the ratio of the maximal growth rates.

*Figure 3—figure supplement 2E* shows the resulting fits of competitive colonization starting with WT and acapsular strains at a 1:1 ratio. Because of the start at a higher concentration than in *Figure 3—figure supplement 2B* and C, the effect of the carrying capacity is felt early and the fit underestimates the net growth rate in the beginning, but there is clearly a lower net growth rate for the acapsular relative to the WT. The second part of the dynamics show a decrease of the acapsular relative to the WT. Fitting the curve gives an estimate of $rr = r_a/r_w$ , but because there are only a few points in this second part, the estimates are noisy. They are found to be between 0.5 and 1.1, compatible with the expected value of 1.

## 3. Competition

### 3.1 General dynamics

*Appendix 1—figure 4* depicts the general dynamics of the competition. The WT and acapsular are thought to interact only through competition for food. After a first bottleneck of survival probability $q_i$ for each initial bacteria, their dynamics consists of

- Fixed loss rate of $c_i$ (with $i$ standing for either WT or acapsular; and the loss rate at least equal to the cecum turnover rate)
- With a maximal shared carrying capacity $K_{max}$ , a growth rate of $r_i \left(1 - \left(W + A\right)/K_{max}\right)$ , with $r_i$ the maximal growth rate, $W$ the WT abundance, and $A$ the acapsular abundance.

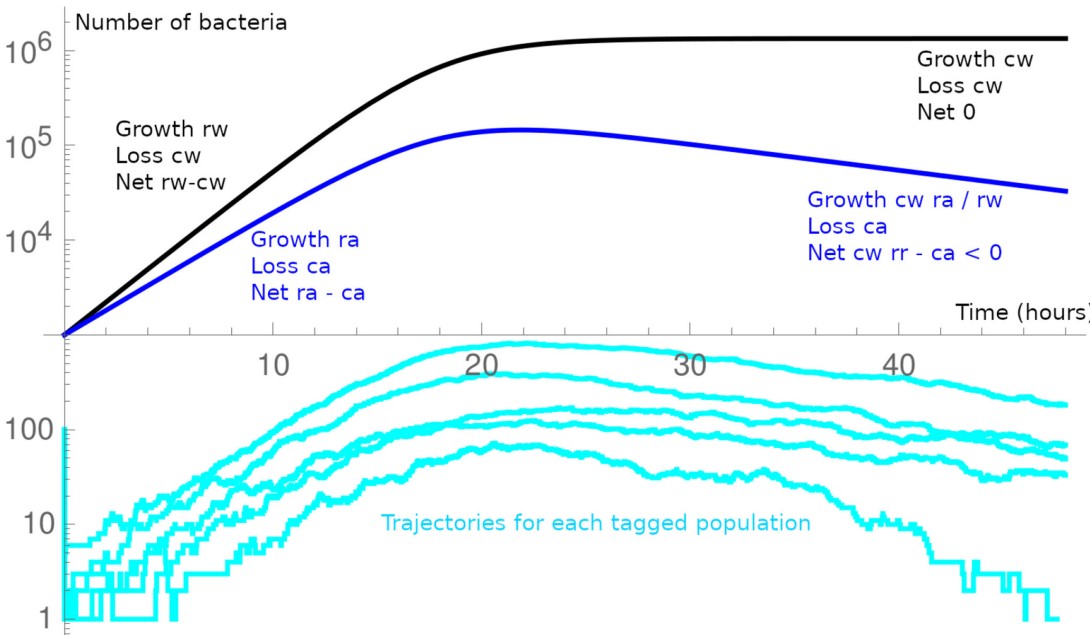

**Appendix 1—figure 4.** Schematic of the general dynamics in the minimal model. The WT (black) and acapsular (blue) population are in competition. After a first bottleneck (at $t = 0$), as the acapsular has a larger loss rate, its net growth rate is smaller. As the bacteria only interact through food, they follow their own dynamics until carrying capacity is reached, then WT is with a null net growth rate, and the acapsular, with its higher loss rate, has a negative net growth rate (we usually assume the absolute growth rate are equal, i.e., $rr = r_a/r_w = 1$). A barcoded population may be lost at the first bottleneck; or in the initial dynamics at low numbers; or towards the end of the experiment when the acapsular population decreases.

The effective carrying capacity for the WT is $K = K_{max} \left(1 - c_w/r_w\right)$ .

Early on, there are few WT and acapsular bacteria, so they grow at their maximal rate. We will approximate the more realistic logistic growth by an exponential growth until reaching the effective carrying capacity. The overall dynamics is very similar to the logistic growth, while being much easier to handle analytically. Then in the experiments we analyze, given that WT and acapsular start in similar abundances, and that the acapsular has a smaller net growth rate, the WT is much more abundant than the acapsular when the effective carrying capacity is reached. Thus, the time to reach carrying capacity is approximately $t_w$ , such that $K = n_{0w}q_w\exp\left(\left(r_w - c_w\right)t_w\right)$ . After $t_w$ , the net

growth rate of the WT is zero, as growth compensate loss, and because $W \gg A$, $W + A \simeq W \simeq K$, thus the net growth rate of the acapsular is

$$net_a' = r_a \left(1 - K/K_{max}\right) - c_a = r_a c_w / r_w + net_a - r_a = net_a - r_a \left(1 - c_w / r_w\right) = net_a - rr net_w. \quad \text{(A38)}$$

If $rr = r_a / r_w$ is close to 1, and because $net_a = r_a - c_a < r_w - c_w = net_w$ , $net_a'$ is predicted to be smaller than 0, and this is in agreement with experimental data showing that the acapsular abundance in competition decreases once the WT has hit carrying capacity.

The generating function for the size of a lineage starting from for an acapsular bacterium is $1 - q_a + q_a g_{up} \left(g_{plat}\left(z\right)\right)$ with $g_{up}$ the generating function for the exponential growth when the population size goes up

$$g_{up}\left(z\right) = 1 - \frac{net_a \left(1 - z\right) \exp\left(net_a t_w\right)}{r_a z - c_a + \left(1 - z\right) r_a \exp\left(net_a t_w\right)}, \quad \text{(A39)}$$

and $g_{plat}$ is the generating function for the acapsular for the phase with a plateau in the WT population size:

$$g_{plat}\left(z\right) = 1 - \frac{net'_a \left(1 - z\right) \exp\left(net'_a \left(t_{tot} - t_w\right)\right)}{r_a \frac{c_w}{r_w} z - c_a + \left(1 - z\right) r_a \frac{c_w}{r_w} \exp\left(net'_a \left(t_{tot} - t_w\right)\right)} \quad \text{(A40)}$$

and thus, the overall survival probability is $q_a \left(1 - g_{up}\left(g_{plat}\left(0\right)\right)\right)$ .

When bacteria are not in competition, they reach a large carrying capacity, so later loss is negligible, resulting in a total survival probability for one bacterium of

$$q_{i,app} = q_i \left(1 - c_i / r_i\right) \quad \text{(A41)}$$

## 3.2 Relative ratio

The relative ratio is $n_{f,WT} n_{0,a} / \left(n_{0,WT} n_{f,a}\right)$ , with $f$ for the final population size, 0 for the number of bacteria in the inoculum, $WT$ for the wild type, and $a$ for the acapsular. We start by its general expression; then show that for $rr = r_a / r_w = 1$, the expression can be simplified and expressed as a function of the parameters measured in other experiments, thus allowing for a prediction; and in the case of $rr \neq 1$ we discuss different expressions.

The expected relative ratio between the WT and the acapsular is

$$R = \frac{q_w \exp\left(net_w t_w\right)}{q_a \exp\left(net_a t_w\right) \exp\left(net'_a \left(t_{tot} - t_w\right)\right)} = \frac{K}{n_{0w} q_a \exp\left(net_a t_w\right) \exp\left(net'_a \left(t_{tot} - t_w\right)\right)} \quad \text{(A42)}$$

$$R = \frac{K}{n_{0w} q_a} \exp\left(-net_a t_w - net'_a \left(t_{tot} - t_w\right)\right) \quad \text{(A43)}$$

Using the expression for $net'_a$ from (38),

$$R = \frac{K}{n_{0w} q_a} \exp\left(-rr net_w t_w - \left(net_a - rr net_w\right) t_{tot}\right) \quad \text{(A44)}$$

With $n_{0w}$ the initial number of WT bacteria, $n_{0w} q_w \exp\left(t_w net_w\right) = K$, leading to

$$R = \frac{q_w^{rr}}{q_a} \left(\frac{K}{n_{0w}}\right)^{1-rr} \exp\left(\left(rr net_w - net_a\right) t_{tot}\right) \quad \text{(A45)}$$

For the Oligo and LCM microbiota, the colonization experiments show that the colonization probability is very similar for the WT and the acapsular, so that it can be assumed that $q_a = q_w$ in these cases. There is no reason a priori for this to be different for GF mice, so this assumption also extends to GF mice. For SPF mice, the apparent colonization probability was quite different, so let us define $qq = q_a / q_w$ . For the colonization process, as the *B. theta* population grows to large sizes, $q_{i,1,app}$ the apparent colonization probability of type $i$ (WT or acapsular) in the colonization process (1 stands for colonization with a single strain type), $q_{i,1,app} \simeq q_i \left(1 - c_i / r_i\right)$ . Then

$$qq_{SPF} = \frac{q_{a,1,app} r_w net_a}{q_{w,1,app} r_a net_w} \quad \text{(A46)}$$

When $\mathbf{rr} = \mathbf{r_a}/\mathbf{r_w} = 1$:

Given that the WT and the acapsular grow at the same rate in vitro, the null assumption is to take them also growing at the same rate in vivo, but with different clearance rates, leading to different net growth rates. With this assumption,

$$R \left( rr = 1 \right) = \frac{q_w}{q_a} \exp \left( \left( net_w - net_a \right) t_{tot} \right) = \frac{1}{qq} \exp \left( \left( net_w - net_a \right) t_{tot} \right) \tag{A47}$$

$qq = 1$ for GF, Oligo and LCM, and is for SPF.

This prediction mixes results from the growth experiments ($net_w$ and $net_a$), and results from the colonization experiment ($qq = 1$ in most cases, $q_{a,1,app}$ and $q_{w,1,app}$ to calculate $qq$ in the SPF case).

**When $rr \neq 1$**

## Predicted relative ratio

Reformulating (45):

$$R = \frac{q_w^{rr-1}}{qq} \left( \frac{K}{n_{0w}} \right)^{1-rr} \exp \left( \left( rrnet_w - net_a \right) t_{tot} \right) . \tag{A48}$$

For GF, Oligo, and LCM mice, $qq$ is taken as = 1, and for SPF, expression (46) is used; for $q_w = q_{w,1,app}/ \left( 1 - c_w/r_w \right) = q_{w,1,app} \left( net_w + c_w \right) /net_w$ . In summary, $qq$ and $q_w$ are taken from the colonization experiments, $net_w$ and $net_a$ from the growth experiments, $n_{0w}$ and $t_{tot}$ are controlled experimental parameters, and $K$ is experimentally measured. As explained earlier, there are indication that $rr$ should be = 1, but in the confidence interval calculations we explore variations around 1.

## Relative ratio given the loss

Reformulating (45):

$$R = \frac{q_a^{rr-1}}{qq^{rr}} \left( \frac{K}{n_{0w}} \right)^{1-rr} \exp \left( \left( rrnet_w - net_a \right) t_{tot} \right) \tag{A49}$$

As before, $qq$ is taken as 1, except for SPF mice, for which expression (46) is used. The difference with the previous expression is that now $q_a$ is taken from the competition experiment. There are two expressions linking $q_a$ to the apparent survival probability $q_{a,app}$ .

- In the limit when the population of acapsular remains high enough at the end of the experiment despite competition with the WT, then the same approximation as in the colonization experiments can be made, and $q_a \simeq q_{a,app}/ \left( 1 - c_a/r_a \right) = q_{a,app} rr \left( net_w + c_w \right) /net_a$ . Note that here one additional assumption is made, on the value of $c_w$ . It is generally taken as the cecum turnover rate, which is its minimal value, but higher values are explored for the confidence intervals.

- The full expression is $q_a = q_{a,app}/ \left( 1 - g_{up} \left( g_{plat} \left( 0 \right) \right) \right)$

## 3.3 Predicted survival probability

The expression for the survival probability is $q_a \left( 1 - g_{up} \left( g_{plat} \left( 0 \right) \right) \right)$

The estimate for $q_a$ is taken from the colonization experiment, with the approximation $q_{a,1,app} = q_a \left( 1 - c_a/r_a \right)$ , thus

$$p_{surv} = q_{a,1,app} \frac{\left( 1 - g_{up} \left( g_{plat} \left( 0 \right) \right) \right)}{1 - c_a/r_a} . \tag{A50}$$

As seen earlier, $g_{up}$ depends on $net_a$ , $t_w$ , $r_a$ , $c_a$ . Also, $g_{plat}$ depends on $net'_a$ , $t_{tot}$ , $t_w$ , $r_a c_w/r_w = rrc_w$ , $c_a$ . We note that $net_a' = net_a - rrnet_w$ , that $r_a = rrr_w = rr \left( net_w + c_w \right)$ and that $c_a = r_a - net_a = rr \left( net_w + c_w \right) - net_a$ . As $n_{0w} q_w \exp \left( t_w net_w \right) = K$ , and $q_w \simeq q_{w,1,app}/ \left( 1 - c_w/r_w \right) = q_{w,1,app} \left( net_w + c_w \right) /net_w$ , $t_w = \log \left( Knet_w/ \left( n_{0w} q_{w,1,app} \left( net_w + c_w \right) \right) \right) /net_w$ .

Overall, the survival probability ends up being dependent on $q_{a,1,app}$ , $q_{w,1,app}$ , (both measured in the colonization experiment), $net_a$ , $net_w$ (both measured in the growth experiment), $t_{tot}$ , $n_{0w}$ (both known controlled parameters of the experiment), $K$ (measured), $c_w$ (in general taken as its lower bound, the cecum turn over), and $rr$ (generally taken as 1).

## 3.4 Effect of a fixed delay in growth

If there was a delay $\tau_i$ for strain $i$ before starting growth after ingestion, then

$$q_{i,1,app} = q_i \exp\left(-c_i \tau_i\right)\left(1 - c_i/r_i\right), \tag{A51}$$

$K = n_{0,w} q_w \exp\left(-c_w \tau_w\right) \exp\left(net_w \left(t_w - \tau_w\right)\right)$ (note that here $t_w$ is the time at which carrying capacity is reached, with duration $\tau_w$ of population decrease, and $t_w - \tau_w$ of net growth $net_w$ of the wildtype), thus

$$\exp\left(net_w \left(t_w - \tau_w\right)\right) = \frac{K \exp\left(c_w \tau_w\right)}{n_{0,w} q_w} = \frac{K net_w}{n_{0,w} q_{w,1,app}\left(net_w + c_w\right)} \tag{A52}$$

As in the model with no delay, we aim to predict the relative ratio and the survival probability, with parameter values taken from other experiments.

The expected relative ratio between the WT and the acapsular is

$$R = \frac{q_w \exp\left(-c_w \tau_w\right) \exp\left(net_w \left(t_w - \tau_w\right)\right)}{q_a \exp\left(-c_a \tau_a\right) \exp\left(net_a \left(t_w - \tau_a\right)\right) \exp\left(net'_a \left(t_{tot} - t_w\right)\right)} \tag{A53}$$

As $net_a' = net_a - rr net_w$ , replacing $\exp\left(net_w \left(t_w - \tau_w\right)\right)$ by previous expression, and after some calculations,

$$R = \frac{1}{qq}\left(\frac{K \exp\left(c_w \tau_w\right)}{n_{0w} q_w} \exp\left(net_w \tau_w\right)\right)^{1-rr} \exp\left(\left(net_w + c_w\right)\left(rr \tau_a - \tau_w\right)\right) \exp\left(\left(rr net_w - net_a\right) t_{tot}\right) \tag{A54}$$

If $rr = 1$, then

$$R = \frac{1}{qq} \exp\left(\left(net_w + c_w\right)\left(\tau_a - \tau_w\right)\right) \exp\left(\left(net_w - net_a\right) t_{tot}\right) \tag{A55}$$

As before, the assumption is $qq = 1$, except for the SPF, when

$$qq = q_{a,1,app}\left(1 - c_w/r_w\right) \exp\left(-c_w \tau_w\right) / \left(q_{w,1,app}\left(1 - c_a/r_a\right) \exp\left(-c_a \tau_a\right)\right)$$

If $rr$ different from 1, in the case of a prediction from the rest of the experiments,

$$R = \frac{1}{qq}\left(\frac{K \exp\left(c_w \tau_w\right)}{n_{0w} q_w} \exp\left(net_w \tau_w\right)\right)^{1-rr} \exp\left(\left(net_w + c_w\right)\left(rr \tau_a - \tau_w\right)\right) \exp\left(\left(rr net_w - net_a\right) t_{tot}\right) \tag{A56}$$

$$R = \frac{1}{qq}\left(\frac{K net_w}{n_{0w} q_{w,1,app}\left(net_w + c_w\right)} \exp\left(net_w \tau_w\right)\right)^{1-rr} \exp\left(\left(net_w + c_w\right)\left(rr \tau_a - \tau_w\right)\right) \exp\left(\left(rr net_w - net_a\right) t_{tot}\right) \tag{A57}$$

If $rr$ is different from 1, in the case of checking for internal consistency of the competition experiment,

$$R = \frac{1}{qq^{rr}}\left(\frac{K \exp\left(c_w \tau_w\right)}{n_{0w} q_a} \exp\left(net_w \tau_w\right)\right)^{1-rr} \exp\left(\left(net_w + c_w\right)\left(rr \tau_a - \tau_w\right)\right) \exp\left(\left(rr net_w - net_a\right) t_{tot}\right) \tag{A58}$$

with $q_a$ is such that $q_{a,app} = q_a \exp\left(-c_a \tau_a\right)\left(1 - g_{up}\left(g_{plat}\left(0\right)\right)\right)$ (note that $g_{up}$ is for $t_{up} = t_w - \tau_w$ , and $g_{plat}$ for $t_{plat} = t_{tot} - t_w$).

And the expected survival probability is $q_a \exp\left(-c_a \tau_a\right)\left(1 - g_{up}\left(g_{plat}\left(0\right)\right)\right)$ , with $q_a \simeq q_{a,1,app} \exp\left(c_a \tau_a\right) / \left(1 - c_a/r_a\right)$ , and thus

$$surv = q_{a,1,app}\left(1 - g_{up}\left(g_{plat}\left(0\right)\right)\right) / \left(1 - c_a/r_a\right) \tag{A59}$$

## 3.5 List of parameters and values

| Symbol | Meaning (more info) | How determined/values taken (germ-free [GF], Oligo, low-complexity microbiota [LCM], specific pathogen free [SPF]) |
|---|---|---|
| $qq$ | $q_a/q_w$ (1) | 1, 1, 1, Depends on model |
| $n_{0w}$ | Number of WT bacteria in the inoculum (2) | Controlled and measured |
| | | $10^{7.63\pm0.1}$, $10^{7.54\pm0.17}$, $10^{7.31\pm0.49}$, $10^{7.56\pm0.20}$, , , |
| $K$ | Effective carrying capacity (3) | Measured |
| | | $10^{10.86\pm0.23}$, $10^{10.17\pm0.47}$, $10^{9.49\pm0.33}$, $10^{7.35\pm0.53}$, , , |
| $rr$ | Ratio of the growth rates, $r_a/r_w$ (4) | 1 [0.5–1.2] (for all microbiota) |
| $net_w$ | WT net growth rate, $r_w - c_w$ (5) | Measured in the growth experiments in Oligo and SPF |
| | | 0.50/h $[0.35 - 0.66]$, 0.50/h $[0.45 - 0.56]$, 0.50/h $[0.45 - 0.56]$, 0.37/h $[0.26 - 0.44]$, , , |
| $net_a$ | Acapsular net growth rate, $r_a - c_a$ (5) | Measured in the growth experiments in Oligo |
| | | 0.45/h $[0.30 - 0.50]$, 0.40/h $[0.35 - 0.44]$, 0.40/h $[0.35 - 0.44]$, 0.27/h $[0.06 - 0.44]$, , , |
| $q_{a,app}$ | Apparent survival probability, | Measured in the competition experiments |
| | acapsular in competition (6) | $10^{-1.27}[-1.35,-1.19]$, $10^{-3.4}[-3.47,-3.33]$, $10^{-2.63}[-2.75,-2.51]$, $10^{-6.59}[-6.64,-6.54]$ |
| $q_{a,1,app}$ | Apparent survival probability (7) | Measured in the colonization experiments (except GF) |
| | when only acapsular | $10^{-0.7}[-1.3,-0.1]$, $10^{-1.48}[-1.65,-1.31]$, $10^{-1.42}[-1.58,-1.26]$, $10^{-3.60}[-3.77,-3.43]$ |
| $q_{w,1,app}$ | Apparent survival probability (7) | Measured in the colonization experiments (except GF) |
| | when only WT | $10^{-0.7}[-1.3,-0.1]$, $10^{-1.54}[-1.71,-1.37]$, $10^{-1.49}[-1.64,-1.34]$, $10^{-2.34}[-2.51,-2.17]$ |
| $tot$ | Total experimental time (8) | Fixed at 44 hr (taken 44 hr [40,48]) |
| $c_w$ | Loss rate of the WT (9) | Minimum is the cecum turnover rate |
| | | 0.13/h $[0.13 - 0.23]$, 0.13/h $[0.13 - 0.23]$, 0.13/h $[0.13 - 0.23]$, 0.23/h $[0.23 - 0.33]$, , , |
| $m_w$ | Mean growth delay for the WT (10) | 3.2 hr [0–5.4] |
| | Delay model only | |
| $m_a$ | Mean growth delay for the acapsular (10) | 7.7 hr [4.0–11.6] |
| | Delay model only | |
| $r_w$ | Growth rate of the WT | $r_w = net_w + c_w$ |
| $r_a$ | Growth rate of the acapsular | $r_a = rr\left(net_w + c_w\right)$ |
| $c_a$ | Loss rate of the acapsular | $c_a = r_a - net_a = rr\left(net_w + c_w\right) - net_a$ |
| $q_i$ | Survival probability initial bottleneck $i$ ($w$ or $a$) (11) | Estimated using $q_{i,1,app}$ or $q_{a,app}$ |
| $t_w$ | Time for the WT to reach carrying capacity | Estimated from the other parameters |

## Detailed notes

1. As the colonization experiment gives very similar apparent probability for WT and acapsular, except for SPF, the assumption is that $qq = 1$ for all except SPF. For SPF, $qq = q_a/q_w$, and in the simple model, taking the approximation that in the colonization experiments, $q_{app} = q\left(1 - c/r\right)$, then $qq = \left(q_{a,app}\left(1 - c_w/r_w\right)\right)/\left(q_{w,app}\left(1 - c_a/r_a\right)\right)$, which can also be written as $= \left(q_{a,1,app}net_w rr\right)/\left(q_{w,1,app}net_a\right)$.

2. Measured in the inoculum, averaged over the inoculum used for the given microbiota, ± the standard deviation between values for the different used inoculum (average and sd calculated on the log values). For GF, only one inoculum is used, so there is no standard deviation, the error is estimated ±0.1 of the log10 of the concentration.

3. Actually, what is measured is the number of bacteria per g of cecum content, whereas $K$ is the absolute value. The assumption is that the cecum is about 1 g. In reality, it is often a bit smaller, but the difference is small compared to the differences in observed final bacterial concentrations.

4. The rationale is that in vitro, WT and acapsular have the same growth rate, so there is no specific reason to believe they are different. So, this ratio is usually taken as 1. This ratio could in principle be determined from the last part of the dynamics of the acapsular in competition with the WT in the growth experiment. Though there are only a few data points, 1 is in the range of fit values, and the minimal and maximal values fit values are used for the confidence interval.

5. We estimated the net growth rate for the acapsular strain in Oligo: $net_a = 0.40/h \, (0.35 - 0.44)$ (± standard deviation). For the WT strain in Oligo, $net_w = 0.50/h \, (0.45 - 0.56)$, and in SPF $0.37/h$ (0.26–0.44, here the lowest and higher fit values). In the experiment shown in *Figure 3—figure supplement 2*, the growth rate of bacteria in Oligo and LCM was similar, so the same values are taken for Oligo and LCM for both acapsular and WT strains. GF mice seem closer in terms of turnover to the LCM and Oligo mice, so the same growth rate is taken for GF for WT strains, increasing the window of uncertainty by ±0.1/hr to consider the data limitation. For the acapsular strain, we assume less clearance based on the lack of competing microbiota, therefore we used an average $net_a = 0.45$. For SPF, it is observed for the WT that the net growth rate is decreased by 0.13/hr, which is consistent with a higher turnover in the cecum of SPF mice (higher by 0.1/hr compared to GF mice), which points towards an increase in $c$ rather than a decrease in $r$. For $net_a$ in SPF, in the absence of direct data, the assumption is thus to take the same decrease relative to the Oligo mice, with a higher uncertainty range to consider the absence of direct data. Note that with this the lowest boundary for the net growth rate of acapsular in SPF is small, and this is consistent with the observations that in some experiments, acapsular in SPF barely grows.

6. Measured from the barcode loss in the competition experiment.

7. Measured in the colonization experiments, the value taken is the estimate via the barcode loss. For GF, a lower bound is the apparent survival probability in the competition experiment ($10^{-1.27} \simeq 10^{-1.3}$), as it is smaller than in the colonization experiments. An upper bound is considering that in the simple model, subsequent survival is at most equal to $(1 - c/r) = net/(net + c)$, and with a net growth rate of the order of 0.53/hr (WT) and 0.40/hr(acapsular) (both measured in Oligo but expected to be similar in GF), and the $c$ at least equal to 0.13/hr, the survival is at most $\simeq 10^{-0.1}$. The main value is taken as the middle (in log scale) between these two boundaries.

8. The initial survival probability may represent very early processes, in the stomach and small intestine, so the time spent in the cecum is actually smaller than 48 hr (the total time between inoculation and the end of the experiment), so it is why in exploring the parameter space the choice is made to take 40–48 hr as the confidence interval.

9. The clearance rate is at least equal to the cecum turnover rate. It is 0.13/hr for the GF, and 0.23/hr for the SPF. LCM and Oligo are thought to be closer to the GF. It would be higher in the presence of killing. Without evidence of killing, the main value is this minimal value. The upper bound is taken as this value, +0.1/hr to represent the impact of potential killing.

10. The delay model explores the possibility that the acapsular takes a longer time to recover and grow back. In the growth curves, all are consistent with exponential growth from 12 hr onward, with points beforehand that suggest some delay, explored in Section 2.5.

11. In the colonization experiments, the bacterial lineages reach and remain at a large population size, so later loss is negligible, thus $q_{i,app} \simeq q_i \, (1 - c_i/r_i)$ for the simpler model. See text for other models.

## 3.6 Computation

The code used for computation is uploaded in the code repository. The results are first computed for the main values of the parameters, for all the models. Then, for a chosen number of iterations (in general 1000), a new set of parameters is taken at random. For each parameter, with probability 0.5 it is chosen randomly uniformly between the lower bound and the main value; and with probability

0.5, it is chosen uniformly between the main value and the upper bound of the confidence interval. It is checked that these parameters combined with $q_{i,1,app}$ lead to $q_a$ and $q_w$ below 1. For the simple model, if the resulting $q_i > 1$, all the parameters are chosen again at random. For the other models, first, for at most 300 iterations, only the parameters linked to the delay are changed, and then if the result gives still survival greater than 1, a full set of parameters is again chosen randomly. This procedure helps avoid bias in the choice of the parameters value.

The results are computed for each of these sets of parameters, and the confidence interval is given as the mean ± the standard deviation on all these iterations.

## 4. Acute challenges

### 4.1 Estimate of $n_0$

We need to estimate the number of barcoded bacteria in the cecum at the start of the challenge from the intermediate data collected in feces, which are a subsampling of the cecum, with potential bias.

### 4.1.1 Cecum mass

A first question is the mass of the cecum as the feces gives only access to a concentration. For all cases, the conditions are like the control conditions up to day 0, thus reasonably at day 0 the cecum is expected to weight like the cecum in the control case at day 3 at the end of the experiment. The average mass of the cecum in GF mice is 2.95 g, standard deviation 0.62 g (*Hoces et al., 2022*).

### 4.1.2 Concentration in feces vs. cecum

At day 3, the data consists of both feces and cecum, thus concentrations can be compared; see *Appendix 1—figure 5*. Using only the data from the untagged strain (less bias because no missing points, but overall there is no large difference between dashed and dotted lines), feces are $10^{0.67+-0.23}$ more concentrated than the cecum for the control (± represent twice the standard deviation for a 95% confidence interval), feces are slightly less concentrated than the cecum for HFD (high-fat diet challenge), and feces are less concentrated than the cecum in Stm (*Salmonella* challenge).

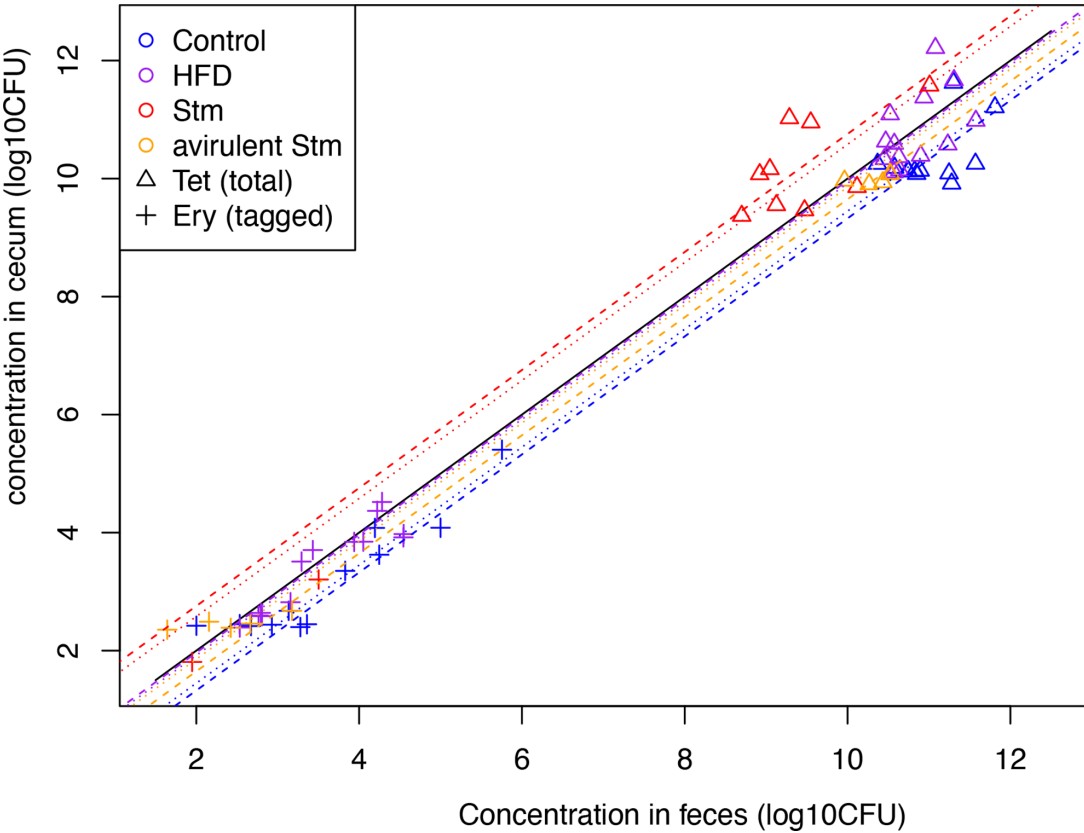

**Appendix 1—figure 5.** Concentration of *B. theta* in the cecum and the feces. Concentration of untagged (triangle) and barcoded (cross) on day 3 after acute challenge (see color legend for each challenge). The dashed lines represent the average ratio considering only the untagged strain data, whereas the dotted lines represent the average of both untagged and barcoded strains (except the case of total loss of barcoded).

### 4.1.3 Factor to convert concentration in feces at day 0 to absolute number in cecum at day 0

To estimate $n_0$, the total number of barcoded bacteria in a mouse at day 0 is taken as the number of erythromycin-resistant colonies in feces, divided by the weight of the sample to obtain the concentration, then multiplied by the mean cecum mass. Finally, the concentration factor for the control case is applied. Then for a given tag, the counts are used to get the proportion of bacteria with that tag in the total number of barcoded bacteria.

### 4.1.4 Tags unseen at day 0

In most cases, tags are both seen in feces at day 0, and in cecum at day 3.

In a fraction of cases, tags are seen in feces at day 0, but not in cecum at day 3. In this case, as the sample is a significant portion of the cecum, it is unlikely that the tag is actually still present, so here we count this situation as a loss.

In the counts of barcoded bacteria, there are a few cases where some tags are not seen at day 0, but still are seen at day 3, due to the feces sample weighting in average less than 10% of the cecum and these barcoded bacteria being in very low numbers.

This also points to the possibility that in the cases where the tags are not seen neither at day 0 nor at day 3, they may have been actually present at day 0 and lost before day 3.

For the loss, we will analyze the data:

- Excluding all cases in which the tags were not seen at day 0.
- Excluding all cases in which the tags were not seen neither at day 0 nor day 3, and for tags not seen at day 0 and seen at day 3, $n_0$ for day 0 is taken as the average estimated total number of barcoded bacteria in that mouse at day 0, divided by 6 (total number of tags).

- For all cases with no tags seen at day 0, $n_0$ for day 0 is taken as the average estimated total number of barcoded subpopulations in that mouse at day 0, divided by 6 (total number of tags).

In practice, the differences in the results are negligible.

## 4.2 Method of the loss

### 4.2.1 Survival probability of a bacterial lineage

We use the same method as in previous section to estimate the colonization probability, except that $n_0$ is estimated from the feces at day 0 instead of the inoculum.

There was no loss for neither the control nor HFD (there is one loss in HFD, but it is also a loss at day = 0, for a large total number of barcoded bacteria in the feces at day 0 in that mouse, so it is likely that this tag was actually lost at day 0). Then to obtain a lower bound for the survival probability, the case with the smallest $n_0$ is taken as lost and the survival probability is calculated from it.

### 4.2.2 Results

- There is no loss for the control case, so the best estimate of $\beta$ is 1, with lower bound $10^{-1.5}$.
- There is no loss for the HFD case, so the best estimate of $\beta$ is 1, with lower bound $10^{-1.75}$.
- For Stm, $\beta = 10^{-3.28 \pm 0.10}$ (for the incertitude: for the log10 of the concentration, the standard error is 0.21, but there are 14 different mice for Stm, thus 0.06 overall expected on the combined data if the ratio of the concentrations are independent in the different mice; for the log10 of the survival probability using a Bayesian approach, the standard deviation is found to be around 0.08; so overall the standard deviation is expected to be around 0.10, then taking 2 standard deviations we find the result).
- For the avirulent Stm, $\beta = 10^{-2.35 \pm 0.3}$ (similar reasoning)

Thus attenuated Stm imposes a relatively strong bottleneck on *B. theta*. The bottleneck from the avirulent *Salmonella* is less stringent. HFD does not have a strong enough effect to be detected in the experimental conditions.

### 4.2.3 Interpretation

This survival probability gives the probability that a bacteria present at day = 0 has still a lineage in the cecum at day 3.

It could either come from a temporary bottleneck or from constant loss during the challenge. If there is a constant turnover rate $c$, then the survival after a time $t$ is $1/(1 + ct)$. For $c$ in GF mice of 0.13/hr, and for $3 \times 24$ hr, we find a survival of about 0.1 $(10^{-1})$, which is compatible with the results for the Control and HFD.

HFD could be causing a larger $c$, but as no loss is observed, we cannot quantify it.

For Stm, the decrease in total population size is at most a factor of 10, whereas survival probability is well below 0.1, pointing to a more stringent bottleneck followed by re-growth (and possibly additional continuous loss).

