## [Editor Report]

The authors have developed an innovative approach to analyze microbial population dynamics within a host and used this technique to address an important question – how the composition of the microbiota influences the intestinal colonization of an encapsulated vs unencapsulated version of an important commensal organism. The revisions add evidence to support their claims and make the approach more accessible.

---

## [Decision Letter]

**Decision letter after peer review:**

[Editors’ note: the authors submitted for reconsideration following the decision after peer review. What follows is the decision letter after the first round of review.]

Thank you for submitting the paper "Fitness advantage of *Bacteroides thetaiotaomicron* capsular polysaccharide is dependent on the resident microbiota" for consideration by *eLife*. Your article has been reviewed by 3 peer reviewers, one of whom is a member of our Board of Reviewing Editors, and the evaluation has been overseen by a Senior Editor. The reviewers have opted to remain anonymous.

Comments to the Authors:

We are sorry to say that, after consultation with the reviewers, we have decided that this work will not be considered further for publication by *eLife* given the extensive amount of experiments and data that would be required to address the major concerns. In general, *eLife* only allows for one revision round with a target of no more than 8 weeks for resubmission. We hope you find the comments and suggestions helpful.

*Reviewer #1 (Recommendations for the authors):*

a. The generation of gnotobiotic mice is not described.

b. Figure 1D cartoon, greys should be 1/7th of population? Better description of inoculum including non-WITS population would be helpful throughout.

c. Figure 1F, the type of average should be expressed in the legend. Similar statements should be made in other figures as well.

d. Figure 1F, Ery appears to have an advantage.

e. Figure 2A cartoon over-simplifies the biology, and suggests all death occurs prior to reaching the site of colonization, when in fact clones can be eliminated in the cecum or flow through without establishing colonization.

f. Figure 3 clearance is misspelled.

g. Figure 4b the challenges appear to independently alter populations in the cecum and feces, suggesting that the feces is not a reliable indicator of cecal populations.

*Reviewer #2 (Recommendations for the authors):*

Line 61 – The use of the term "gnotobiotic" is incorrect, as it does not refer to any specific level of diversity. Gnotobiotics just means that the composition is known, which can be 0, 1 or 100s of microbes.

Need to be careful not to use "CFU" to refer to qPCR data, which measures DNA levels not viable cells. A common technical term is "genome equivalents".

The growth curve format and analysis was unclear. For example, Figure 1b would be aided by overlapping all the lines on the same graph, removing the log scaling on the y axis, and quantifying carrying capacity and lag phase parameters. Error bars are also missing/not visible.

Figure 1e – I cannot see the error bars, only 1 dot/tag is visible.

The correct abbreviation for overnight is "O/N" not "ON". Better yet, just spell out the words to avoid confusion.

Line 112 – this seems like a big assumption – how do we know selection can't occur in 48 hours?

Line 117 – waiting to grow in the cecum also seems like a big assumption to me.

Figure 2c – this panel is really unclear, need to indicate which tags are which by color.

Figures 2d-g – it would help to include the wt and mutant strains on the same graph, to enable comparisons between strains and appropriate statistics.

Figure 2b – multiple typos: "extintino", "optimun"

Figure 3e,f – these curves raised a lot of questions. The first timepoint has high levels, especially for panel f. A time zero needs to be included to rule out background amplification. Given the 3 OOM difference between the two panels in starting levels I'm not sure it is fair to compare the curves. The n is very low here, need more mice and a replicate experiment. Controls should be included with the individual members of the OligoMM community to ensure that there is no background amplification.

Line 257 – this is interesting but seems very speculative. It would really help to formulate a testable hypothesis.

Line 260 – this section did not follow from the rest of the paper and might be better to move it to a follow-up study. The experiments are also very preliminary and conflict with prior literature. For example, the diet experiment would need to be repeated with a matched control diet, 16S profiling to ensure that the rest of the gut microbiota is altered, bile acid quantification, and a longer time series (3 days is quite short).

Figure 4b – the difference in the control group is worrisome, although perhaps that's due to comparing cecal contents to feces. It would probably be safer to just compare the same sample type across groups.

Line 314 – this conclusion is not well supported given all the potential host differences.

*Reviewer #3 (Recommendations for the authors):*

My comments below are mainly focused towards the interpretation of their data, in particular the spatial localization question I bring up below.

– Line 97-100: they state that the antibiotic cassette isn't really altering fitness, which I found surprising – there is generally some fitness cost to antibiotic cassettes, at least in in vitro experiments. Upon examination of Figure 1F, it actually looks like for the wild-type, the Ery cassette wins out in competition. This should at least be tested statistically, as a half-log-fold after 2 days is fairly important biologically.

– Looks like there are consistent biases for certain barcodes in Figure 1E? e.g., pink/purple is consistently lowest in WT, albeit not acapsular. Again, a half-log-fold after 2 days is not small here and would accumulate over time. Can they rule out fitness differences in any other way? I realize that they test for mutation in selection by reinoculating isolated strains, but this does not test for pre-existing mutation.

– Line 171-174: they conclude that size of open niche does not translate linearly into colonization probability, but this seems like a given, since the barcodes are competing with themselves, and so if one barcode happens to have a delay, then the niche that it experiences is now smaller since the other barcodes have expanded to fill part of the overall niche. As a result, I don't think I agree with their conclusion that "net growth during the first 48h post-inoculation has to be lower than in the gnotobiotic mice."

– Figure 3E: It would be helpful to have more information on how they estimated growth rate; I don't see in the plot such a large difference (25%) between 0.53 and 0.4.

– For their conclusions related to increased clearance rate, how do they balance this conclusion/model with one related to variable growth rate in the gut due to nutrient availability and/or spatial localization. I think in particular since they are dealing with colonization, the spatial question is v. important.

[Editors’ note: further revisions were suggested prior to acceptance, as described below.]

Thank you for resubmitting your work entitled "Fitness advantage of *Bacteroides thetaiotaomicron* capsular polysaccharide is dependent on the resident microbiota" for further consideration by *eLife*. Your revised article has been evaluated by Wendy Garrett (Senior Editor) and a Reviewing Editor.

The authors have made major efforts revising the manuscript, substantially addressing the reviewers concerns through a combination of additional experiments and analyses.

A few issues remain to be addressed:

1) There have been significant improvements to the lag phase analysis, and we accept the authors conclusions given their careful language, consideration of alternate hypothesis, and the good fit of their models. However, there continues to be issues in what has become Figure 3 -supplement 2. There appears to be data points in figure supplement 2B that are missing from figure supplement 2D. Perhaps these points are obscured, as occurred previously, but the source data for this supplement has less animals than reported in the legend.

2) Addition of challenge with avirulent Stm is a good experiment to demonstrate that the bottleneck following Stm infection occurs due to inflammation. However, the presentation of the Stm avirulent data in Figure 4C and its discussion in the text needs clarification. In Figure 4 —figure supplement 1 at n0=100 (the only dose used for Stm avirulent), the data for Stm avirulent appear nearly identical to the matched control. Yet, in Figure 4C the bottleneck caused by Stm avirulent appears substantially different than Control and in the text the authors claim that Stm avirulent creates a bottleneck. The authors should clarify why the difference between the two groups is so apparent in Figure 4C, but not as much in Figure 4—figure supplement 1.

3) The authors present three potential scenarios to explain why there is a bottleneck for the acapsular strain in mixed infection: 1) competition prior to reaching the cecum, 2) competition in the cecum prior to growth, and 3) difference in growth kinetics. The authors then carefully explain why they believe the bottleneck is attributable to growth kinetics. However, in the Results section (line 317), the authors propose the scenarios as 1) lag phase, 2) killing in the stomach, and 3) retention in the small intestine. We encourage the authors to include the explanation of why they exclude hypothesis 2, which they mention in the rebuttal, in the main text. Perhaps, a figure showing "a similar delay in CFU arrival in the feces for acapsular B. theta on single colonization as in competition", as they mention in the rebuttal, would help clarify this point.

---

## [Author Response]

[Editors’ note: The authors appealed the original decision. What follows is the authors’ response to the first round of review.]

Reviewer #1 (Recommendations for the authors):a. The generation of gnotobiotic mice is not described.

All gnotobiotic colonies were bred with the listed microbiota. This is now clearly described in the methods section and 16S sequencing data of all microbiotas has been added (Figure 2 —figure supplement 1).

b. Figure 1D cartoon, greys should be 1/7th of population? Better description of inoculum including non-WITS population would be helpful throughout.

We agree there was lack of clarity in description of our inocula throughout the manuscript. In fact, the diagram is correct: The six WITS strains were pooled at equal proportions, then the WITS pool was mixed 1:1 with the untagged population. To address this, we have added the actual measured composition of all inocula either to the main figures or as supplementary figures where several inocula contributed to one graph. We have also improved the methods section describing *B. theta* growth and preparation.

c. Figure 1F, the type of average should be expressed in the legend. Similar statements should be made in other figures as well.

Better information on the averages used and statistics has been added to all figure legends. Apologies for this omission.

d. Figure 1F, Ery appears to have an advantage.

Small differences due to antibiotic resistance cassettes are unavoidable. This statement has been corrected in the text. Additionally, we have run a simulation of the expected effect of fitness contributions due to the antibiotic resistance cassettes on the outcome of the competition experiments and this is expected to be less than 2-fold. As the CI between WT and acapsular *B. theta* range from 20- 10^6^-fold, this is a relatively small error. Additionally, competition experiments were carried out with the antibiotic resistances exchanges with very similar results.

e. Figure 2A cartoon over-simplifies the biology, and suggests all death occurs prior to reaching the site of colonization, when in fact clones can be eliminated in the cecum or flow through without establishing colonization.

Indeed, we are aware that this is an oversimplification of a complex dynamic process, but the cartoon means to exemplify the simplified mathematical model which in the first instance used in Figure 2 assumes no further loss of tags after cecum arrival. This model was initially applied to calculate bottleneck size. This is later complemented with a more complex model depicted in Figure 3. This is now referred to as our “initial” model to avoid further confusion.

f. Figure 3 clearance is misspelled.

Corrected.

g. Figure 4b the challenges appear to independently alter populations in the cecum and feces, suggesting that the feces is not a reliable indicator of cecal populations.

This is a broad issue in microbiome research – by definition feces are always a sub-sample of the end-point of what is going on in the proximal large intestine and has been investigated both in mice and humans. However, it is not ethically justified in this case to sequentially sample the cecum of one mouse, making feces a useful if imperfect read-out for longitudinal sampling. Based on this, we relied on fecal population only to report a pre-challenge state (for which most mice showed all tag strains present). We provided the post-challenge fecal population as a point of comparison to pre-challenge, but all survival probabilities were calculated on based on the lost of tags in the cecal population. This has been clarified in the text and methods (highlighted in the text).

Reviewer #2 (Recommendations for the authors):Line 61 – The use of the term "gnotobiotic" is incorrect, as it does not refer to any specific level of diversity. Gnotobiotics just means that the composition is known, which can be 0, 1 or 100s of microbes.

A good point – we have modified this to “low-complexity” or to simply naming the consortia.

Need to be careful not to use "CFU" to refer to qPCR data, which measures DNA levels not viable cells. A common technical term is "genome equivalents".

Note that we are actually reporting CFU. This was poorly described in the first version of the manuscript and has been much better reported here. qPCR is used to generate relative frequencies of barcoded strains, whilst the total CFU/g of all barcoded strains is determined by plating. The CFU/g of each barcoded strain is then calculated by simple multiplication.

The growth curve format and analysis was unclear. For example, Figure 1b would be aided by overlapping all the lines on the same graph, removing the log scaling on the y axis, and quantifying carrying capacity and lag phase parameters. Error bars are also missing/not visible.Figure 1e – I cannot see the error bars, only 1 dot/tag is visible.

This is an excellent suggestion and has been carried out as suggested. Error bars are present but very narrow.

The correct abbreviation for overnight is "O/N" not "ON". Better yet, just spell out the words to avoid confusion.

Point taken! This has been corrected.

Line 112 – this seems like a big assumption – how do we know selection can't occur in 48 hours?

This is very valid criticism – rather the probability of non-random selection increases over time. This statement has been adapted to be more accurate. By re-isolating clones from the 48h-colonized mice and using these to produce a new barcoded inoculum we could experimentally exclude major increases in fitness over this timeframe in our models (Data in Figure 2 —figure supplement 6).

Line 117 – waiting to grow in the cecum also seems like a big assumption to me.

This statement is based on data that there is little to no colonization of the small intestine by *B. theta.* We have now added data where OligoMM12 were colonized for 8h with a large inoculum to allow consistent detection and we can demonstrate very low colonization of the upper or lower small intestine, consistent with this environment not being permissive for *B. theta* growth (Figure 3 —figure supplement 4).

Figure 2c – this panel is really unclear, need to indicate which tags are which by color.

This is a misunderstanding – each dot is one mouse not one tag (the evenness is based on the complete tag distribution within each animal), now better indicated in the figure legend. For observing the distribution of individual tags on each experiment we have added Figure 2 —figure supplement 3 and 4 including all raw data.

Figures 2d-g – it would help to include the wt and mutant strains on the same graph, to enable comparisons between strains and appropriate statistics.

We consider that separation between the two graphs helps to deliver the message about comparisons for a particular strain (WT or acapsular) among the 3 different microbiotas. For easiness of comparison between strains we have maintain the same y axis range.

Figure 2b – multiple typos: "extintino", "optimun"

Corrected.

Figure 3e,f – these curves raised a lot of questions. The first timepoint has high levels, especially for panel f. A time zero needs to be included to rule out background amplification. Given the 3 OOM difference between the two panels in starting levels I'm not sure it is fair to compare the curves. The n is very low here, need more mice and a replicate experiment. Controls should be included with the individual members of the OligoMM community to ensure that there is no background amplification.

N has been increased to at least 7 mice for these growth curves. Note that there is no qPCR in this panel – the data is from plating and no colonies are detected on the selective antibiotics used from the OligoMM12 colony at time zero – this control is explicitly stated in the methods section referring to this figure. These graphs and their analysis have been updated and are now shown in figure 3 —figure supplement 2

Line 257 – this is interesting but seems very speculative. It would really help to formulate a testable hypothesis.

Based on improved estimates for in vivo growth curves, this section has been considerably modified and I hope is now clearer.

Line 260 – this section did not follow from the rest of the paper and might be better to move it to a follow-up study. The experiments are also very preliminary and conflict with prior literature. For example, the diet experiment would need to be repeated with a matched control diet, 16S profiling to ensure that the rest of the gut microbiota is altered, bile acid quantification, and a longer time series (3 days is quite short).

This is a very fair criticism, and these data were included to give the research community some insight into the limits and possibilities of this approach, rather than to reveal deep biological insight. Given that the phenotype in high-fat diet is below detection limit, and the evenness cannot increase, we would not suggest using a control diet here. This section has been party re-written. We have also included an addition control with colonization with “avirulent” *Salmonella* to clarify the mechanism driving a tight bottleneck in the attenuated *Salmonella* infection. Data on the extent of inflammation is also now included.

Figure 4b – the difference in the control group is worrisome, although perhaps that's due to comparing cecal contents to feces. It would probably be safer to just compare the same sample type across groups.

We completely agree with this observation and note that a large number of publications in this field rely entirely on feces analysis. Owing to this, we have done all the quantitative analyses on cecum content between groups (clarified also in the text) and include the feces analysis only for absolute transparency, and for comparison to published feces-based studies. Specifically, the data points in Figure 4C are based on cecum content.

Line 314 – this conclusion is not well supported given all the potential host differences.

We agree that currently we do not present sufficient data to completely delineate host and microbiota effects. Additional data has been added allowing us to exclude effects of intestinal IgA and of lytic phage, as well as to determine that SPF-microbiota-driven mechanisms are already present 24h post re-colonization of germ-free mice. Again, this text has been extensively re-written.

Reviewer #3 (Recommendations for the authors):My comments below are mainly focused towards the interpretation of their data, in particular the spatial localization question I bring up below.– Line 97-100: they state that the antibiotic cassette isn't really altering fitness, which I found surprising – there is generally some fitness cost to antibiotic cassettes, at least in in vitro experiments. Upon examination of Figure 1F, it actually looks like for the wild-type, the Ery cassette wins out in competition. This should at least be tested statistically, as a half-log-fold after 2 days is fairly important biologically.

We agree that there are inevitable and variable costs of antibiotic cassette production, and these may have a numerical effect on our calculations. To address this, we have carried out a simulation of the effect of a slight fitness advantage/disadvantage of our strains in the competition experiments and come to the conclusion that only a 2-fold difference in the competitive index would result. Given that we have competitive indexes in the rage of 20-10^6^, this is not going to change our conclusions. Nevertheless, this now carefully discussed in the Results section. For our barcode analysis the effect is uniform over all barcodes and therefore does not alter the competition between barcoded strains

– Looks like there are consistent biases for certain barcodes in Figure 1E? e.g., pink/purple is consistently lowest in WT, albeit not acapsular. Again, a half-log-fold after 2 days is not small here and would accumulate over time. Can they rule out fitness differences in any other way? I realize that they test for mutation in selection by reinoculating isolated strains, but this does not test for pre-existing mutation.

“Neutrality” is absolutely the most challenging part of neutral tagging approaches, particularly where strain construction is a multi-step process. To address this, we have:

1) Whole-genome sequenced all of our barcoded strains. This reveals random SNPs in some of the barcoded strains. However, none of these are in genes known to have a major fitness effect in vivo*.*

2) We have re-done our estimates of β sequentially leaving out one of the barcoded strains from the analysis. This evaluation is now shown in Figure 2 supplement 6. Removing any individual barcode had no significant effect on the estimated values of β.

– Line 171-174: they conclude that size of open niche does not translate linearly into colonization probability, but this seems like a given, since the barcodes are competing with themselves, and so if one barcode happens to have a delay, then the niche that it experiences is now smaller since the other barcodes have expanded to fill part of the overall niche. As a result, I don't think I agree with their conclusion that "net growth during the first 48h post-inoculation has to be lower than in the gnotobiotic mice."

This is an interesting point. In the SPF mice, the expansion of the population relative to the inoculum is 100-fold less than in low-complexity microbiota mice, but the probability of an individual clone to colonize is only 10-fold lower. Our logic was as follows: Β multiplied by the total population size of *B. theta* in the inoculum can be interpreted that the number of clones from the inoculum represented in the 48h cecal population. This is 10-fold smaller in SPF mice than in an OligoMM12-colonized mouse. However, the total population size is 100-fold smaller, indicating that the net expansion of each clone required to generate the final population is less. We are using “net growth rate” to mean growth rate minus clearance rate, i.e., equivalent to the rate of expansion of the population.

As the barcoded strains are massively outnumbered by the untagged *B. theta* (between 1000- and 1’000’000-fold depending on the experiment) then we assume that competition mainly occurs between a tagged clone and untagged clones, and this will be the case in both the SPF microbiota background and the low-complexity microbiota. Large stochastic differences in lag phase exist between barcoded clones can have interesting effects on the estimate of β, mainly by increasing the variation in situations where uniform recovery is expected and we did investigate this in the context of some of the models, but the effect on the final estimates of β are small unless there are consistent differences in lag phase distribution between the strains (see the comparison for lag-phase effects for acapsular and wildtype *B. theta*).

– Figure 3E: It would be helpful to have more information on how they estimated growth rate; I don't see in the plot such a large difference (25%) between 0.53 and 0.4.

This data is now added in Figure 3 —figure supplement 2. Indeed, this was a point of considerable discussion between the mathematical modelers and biologists on this project and we have added more mice in order to improve these estimates.

– For their conclusions related to increased clearance rate, how do they balance this conclusion/model with one related to variable growth rate in the gut due to nutrient availability and/or spatial localization. I think in particular since they are dealing with colonization, the spatial question is v. important.

We agree there are major hypotheses relating to differences in growth niches and distributions of growth rates for *Bacteroides* strains in the gut (Lee et al., 2013). The effect of subpopulations with different growth rates was investigated in a previous modeling project looking at *Salmonella* gut colonization (Maier et al., 2014) but at the resolution we are working at, this generally fails to explain the distribution of barcodes recovered. A discussion will be added to the modelling supplement, which will also be re-worded to increase clarity. Of note the cecum does not contain structured mucus layers – these are only present in the mid-distal colon after the point of sampling for β estimates in our work.

[Editors’ note: what follows is the authors’ response to the second round of review.]

The authors have made major efforts revising the manuscript, substantially addressing the reviewers concerns through a combination of additional experiments and analyses.A few issues remain to be addressed:1) There have been significant improvements to the lag phase analysis, and we accept the authors conclusions given their careful language, consideration of alternate hypothesis, and the good fit of their models. However, there continues to be issues in what has become Figure 3 -supplement 2. There appears to be data points in figure supplement 2B that are missing from figure supplement 2D. Perhaps these points are obscured, as occurred previously, but the source data for this supplement has less animals than reported in the legend.

Yikes! Thank you for spotting this error! Figure 3 —figure supplement 2 has been corrected accordingly. We discovered an error in the code that generated Figure 3 – Supplement 2D which had resulted in some data points failing to appear on the graph and source data file. This has been resolved for the source data file and Figure 3- Supplement 2D has been updated. It now includes all points from Figure 3 —figure supplement 2B and C. Additionally, points have been jittered in the x and y-axes in all figures to allow better visualization of overlayed data. Figure legend have also been updated to include describe changes. This does not change the interpretation of the data.

2) Addition of challenge with avirulent Stm is a good experiment to demonstrate that the bottleneck following Stm infection occurs due to inflammation. However, the presentation of the Stm avirulent data in Figure 4C and its discussion in the text needs clarification. In Figure 4 —figure supplement 1 at n0=100 (the only dose used for Stm avirulent), the data for Stm avirulent appear nearly identical to the matched control. Yet, in Figure 4C the bottleneck caused by Stm avirulent appears substantially different than Control and in the text the authors claim that Stm avirulent creates a bottleneck. The authors should clarify why the difference between the two groups is so apparent in Figure 4C, but not as much in Figure 4—figure supplement 1.

We absolutely agree that this is confusing. The reason for the discrepancy is that the qPCR data here was carried out on enrichments of 50% of the cecum, while CFU determination was carried out with much less material so the detection limit for CFU is higher than that for qPCR detection. Figure 4 —figure supplement 1 now shows raw CT values rather than calculated CFUs of each tagged strain: These are the data used in calculation of the model. Here it is clear that the CT values are slightly lower (i.e. effectively all barcodes are present) in the control mice at day 3, whilst some barcodes are below detection (CT>30) in the avirulent *Salmonella*-infected mice. Noticeably, the avirulent *Salmonella* has much less of a bottleneck effect than the attenuated-*Salmonella* infected mice, which had measurable intestinal inflammation: here there is very little barcode recovery (i.e., except in one mouse, the majority of CT values are >30). We hope this is now clearer.

3) The authors present three potential scenarios to explain why there is a bottleneck for the acapsular strain in mixed infection: 1) competition prior to reaching the cecum, 2) competition in the cecum prior to growth, and 3) difference in growth kinetics. The authors then carefully explain why they believe the bottleneck is attributable to growth kinetics. However, in the Results section (line 317), the authors propose the scenarios as 1) lag phase, 2) killing in the stomach, and 3) retention in the small intestine. We encourage the authors to include the explanation of why they exclude hypothesis 2, which they mention in the rebuttal, in the main text. Perhaps, a figure showing "a similar delay in CFU arrival in the feces for acapsular B. theta on single colonization as in competition", as they mention in the rebuttal, would help clarify this point.

This is an excellent point. We have modified the text around line 317 to include the very clear argumentation used in the rebuttal letter. Additionally, one graphs showing the competitive colonization dynamics of both strains have been moved from the mathematical modeling appendix into Figure 3 —figure supplement 2E. Using this data, a new graph has been generated (Figure 3 —figure supplement 2F) in order to better support these claims by showing no difference in the delay during single colonization as in competition.